# The APOE isoforms differentially shape the transcriptomic and epigenomic landscapes of human microglia xenografted into a mouse model of Alzheimer's disease

Kitty B. Murphy[1,2,3], Di Hu[1,2], Leen Wolfs[4,5], Susan K. Rohde ●[4,5], Gonzalo Leguía Fauró[6,7], Ivana Geric[4,5], Renzo Mancuso[6,7], Bart De Strooper ●[4,5,8] & Sarah J. Marzi ●[2,3,9] ✉

Microglia play a key role in the response to amyloid beta in Alzheimer's disease (AD). In this context, the major transcriptional response of microglia is the upregulation of *APOE*, the strongest late-onset AD risk gene. Of its three iso-forms, APOE2 is thought to be protective, while APOE4 increases AD risk. We hypothesised that the isoforms change gene regulatory patterns that link back to biological function by shaping microglial transcriptomic and chromatin landscapes. We use RNA- and ATAC-sequencing to profile gene expression and chromatin accessibility of human microglia xenotransplantated into the brains of male APP$^{NL-G-F}$ mice. We identify widespread transcriptomic and epigenomic differences which are dependent on *APOE* genotype and are corroborated across the profiling assays. Our results indicate that impaired microglial pro-liferation, migration and immune responses may contribute to the increased risk for late-onset AD in *APOE4* carriers, while increased phagocytic capabilities and DNA-binding of the vitamin D receptor in *APOE2* microglia may contribute to the isoform's protective role.

Microglia are key players implicated in the genetic susceptibility and progression of Alzheimer's disease (AD). AD genetic risk pre-dominantly falls within regulatory regions of the genome, including those marked by H3K27ac[1], an epigenetic modification found at active enhancers and promoters[2]. H3K27ac is dysregulated in the brains of individuals with AD[3,4], and microglial H3K27ac regions are strongly enriched for AD genetic risk[5,6]. Microglia have also been associated with AD risk in their open chromatin regions[7–10], and at the level of their transcriptome[11–13]. Recent research using single-cell transcriptomics has highlighted that microglia occur in various distinct subtypes and activation states, which are anticipated to exhibit different epigenomic and transcriptomic responses dependent on their environmental niche. This variation is expected to give rise to different downstream effects on AD pathogenesis. Supporting this is the continued char-acterisation of different microglial phenotypes in AD, including those responsive to amyloid beta (Aβ) aggregates, a pathological hallmark of AD[14–16].

In mouse models, the strongest transcriptional response of microglia to Aβ aggregates is the upregulation of the gene *APOE*[14,17,18], which harbours the strongest genetic risk factor for late-onset AD.

[1]UK Dementia Research Institute at Imperial College London, London, UK. [2]Department of Brain Sciences, Imperial College London, London, UK. [3]Department of Basic and Clinical Neuroscience, Institute of Psychiatry, Psychology and Neuroscience, King's College London, London, UK. [4]VIB Center for Brain & Disease Research, VIB, Leuven, Belgium. [5]Department of Neurosciences and Leuven Brain Institute, KU Leuven, Leuven, Belgium. [6]VIB Center for Molecular Neu-rology, VIB, Antwerp, Belgium. [7]Department of Biomedical Sciences, University of Antwerp, Antwerp, Belgium. [8]UK Dementia Research Institute at University College London, London, UK. [9]UK Dementia Research Institute at King's College London, London, UK. ✉e-mail: sarah.marzi@kcl.ac.uk

*APOE* is involved in regulating cholesterol and other lipid transport across cells[19]. In the periphery, it is produced by macrophages in the liver, while in the brain, it is primarily produced by astrocytes[19]. In humans, APOE has uniquely evolved into three different isoforms encoded by the alleles: *APOE2*, *APOE3*, and *APOE4*. In AD, *APOE2* is thought to be protective, while *APOE4* increases disease risk up to 12-fold in homozygous individuals of certain human populations[20]. Several studies have begun to explore the role of microglial APOE in AD, using different samples and methodologies. In mouse models of AD, mice expressing the APOE4 isoform exhibit a higher abundance of microglia stress and inflammatory markers, a phenotype also observed in human tissue[15]. Furthermore, APOE4 microglia are linked to dysregulated lipid metabolism[21,22], followed by tau phosphorylation[22]. Regarding immune responses, APOE4 microglia induce the signalling of transforming growth factor-β (TGF-β), a multifunctional cytokine[23]. Clearly, APOE plays a critical role in regulating microglia in response to AD pathology. Thus far, studies have predominantly focused on APOE4, but it is equally important to determine the role of APOE2, which may potentially be antagonistic. We hypothesised that different APOE isoforms would differentially regulate the microglial phenotype in response to Aβ pathology. However, the unique evolution of APOE in humans makes it difficult to faithfully recapitulate its effects on AD risk and pathogenesis in mouse models. As previously suggested, investigating the microglial response in human tissue is challenging due to technical limitations and inconsistencies in biological findings[24].

To tackle these challenges, researchers have developed a human microglia xenotransplantation model[25,26], in which iPSC-derived human microglia are xenografted into the brains of mice. Single-cell profiling of these microglia has identified known and novel amyloid-responsive states[27]. These microglial states were enriched for different subsets of AD genetic risk genes, highlighting that multiple microglial states are influenced by AD genetic susceptibility. In addition, shift into what is thought to be a protective and human-specific microglial state was impaired in *APOE4* microglia[16]. In addition to demonstrating the usefulness of this model in disentangling the microglial response to Aβ pathology, this highlights the need to investigate the functional role of AD genetic risk factors in a cell type-specific manner.

Here, we used ATAC-seq and RNA-seq to profile human microglia expressing the different APOE isoforms, which were xenotransplanted into the *App*^NL-G-F mouse model of AD. This enabled us to delineate the effects of the different APOE isoforms on the epigenomic and transcriptomic landscapes of microglia in AD.

## Results

We transplanted iPSC-derived human microglia *APOE2*/0, *APOE3*/0, *APOE4*/0 and an *APOE* knockout (*APOE*-KO) into the brains of the *App*^NL-G-F mouse model of Alzheimer's disease[28]. At 12 months, by which point Aβ pathology is extensive[18,28], microglia were isolated by FACS using human microglia-specific antibodies (CD11b + hCD45+, Fig. 1a). This approach results in a scenario where the manipulations of the *APOE* genotype are restricted to microglia, thereby allowing us to study microglia-autonomous effects of the isoforms. To characterise the epigenomic and transcriptomic landscapes of these microglia, they were profiled using ATAC-seq for open chromatin and RNA-seq for gene expression, respectively. After quality control and pre-processing, we obtained high-quality chromatin accessibility data across 16 mice (*APOE2* = 5, *APOE3* = 5, *APOE4* = 4, *APOE*-KO = 2; Supplementary Data 1, Supplementary Fig. 1) and high-quality transcriptomic data across 17 mice (*APOE2* = 5, *APOE3* = 4, *APOE4* = 5, *APOE*-KO = 3; Supplementary Data 1, Supplementary Fig. 1). Overall, we observed widespread differences in gene expression and chromatin accessibility across microglia of the different APOE isoforms, highlighting the complexity of the microglial response to Aβ pathology. In support of

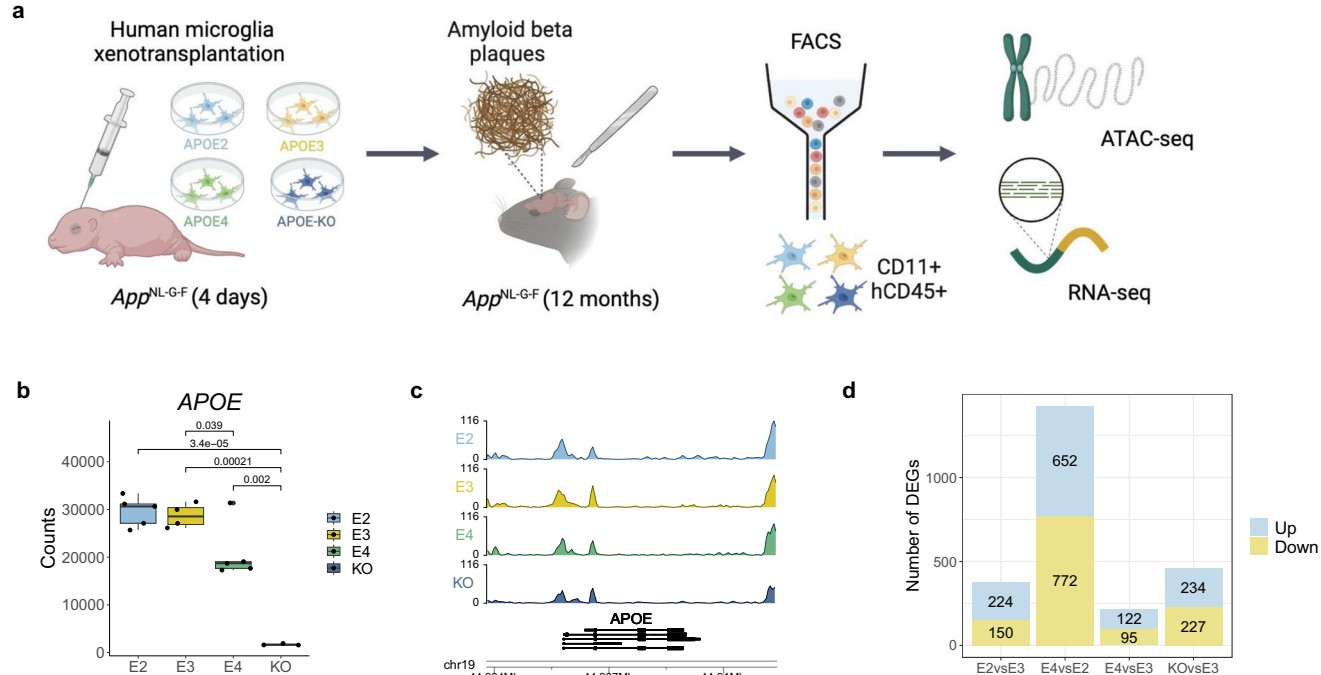

**Fig. 1 | Transcriptomic and epigenomic profiling of xenotransplanted microglia reveals changes to their regulation in Alzheimer's disease across the different APOE isoforms. a** Experimental design for xenotransplantation of iPSC-derived human microglia into the brains of *App*^NL-G-F mice, followed by ATAC-seq (*APOE2* = 5, *APOE3* = 5, E4 = 4, *APOE*-KO = 2) and RNA-seq (*APOE2* = 5, *APOE3* = 4, *APOE4* = 5, *APOE*-KO = 3). Created in BioRender. Marzi, S. (2025) https://BioRender.com/1awx14m. **b** Boxplot of expression profiles of *APOE*, confirming the knockout for 3 out of 5 *APOE*-KO samples. The two-sided Wilcoxon rank-sum test was used to calculate *p*-values. The central mark and edges indicate the 50th (median), 25th and 75th percentiles. Whiskers correspond to 1.5 * the interquartile range (IQR). **c** Genome tracks showing chromatin accessibility signals of all *APOE* alleles around the *APOE* locus. **d** Stacked barplot of the number of differentially expressed genes (DEGs; FDR < 0.05) identified through pairwise comparisons across the experimental groups. Source data are provided as a Source Data file.

the opposing roles of APOE2 and APOE4 in AD risk, the greatest differences were observed between these isoforms.

## APOE isoforms are associated with consistent differences in gene expression and chromatin accessibility

*APOE* expression was significantly lower in three out of five knockout samples (Supplementary Fig. 2), and only these were used in downstream analyses (Fig. 1b). Additionally, *APOE* expression was lower in the *APOE4* microglia compared to *APOE2* and *APOE3*. This is in agreement with previous studies investigating the *APOE* alleles in microglia and astrocytes[29,30]. Chromatin accessibility around the transcriptional start site (TSS) of *APOE* was consistent across the *APOE* groups (Fig. 1c). As the most commonly expressed allele[20], we used *APOE3* as the baseline for pairwise comparisons in the differential expression and chromatin accessibility analyses performed using DESeq2[31]. In addition, we compared the microglia expressing *APOE4* with those expressing *APOE2*. When compared to *APOE3* microglia, differential expression analysis revealed 470 (282 up, 188 down, (FDR < 0.05); Figs. 1d, 2a, Supplementary Data 2) and 332 (170 up, 158 down, (FDR < 0.05); Figs. 1d, 2b, Supplementary Data 2) differentially expressed genes (DEGs) in *APOE2* and *APOE4*, respectively. As expected, given their postulated opposing roles in AD risk, the direct comparison of *APOE4* with *APOE2* revealed the most differences, with 1639 DEGs (751 up, 888 down, (FDR < 0.05; Figs. 1d, 2c, Supplementary Data 2). 126 genes were upregulated and 206 were downregulated in the *APOE*-KO when compared with the *APOE3* isoform (Supplementary Fig. 3, Supplementary Data 2).

Comparison with the *APOE*-KO enabled us to infer whether the transcriptional mechanisms underlying *APOE2* and *APOE4* microglia can be explained by loss and/or gain of function. We used Rank-Rank Hypergeometric Overlap (RRHO) to quantify the degree of overlap between expression signatures in the *APOE*-KO vs *APOE3* and *APOE2* vs *APOE3*, and the *APOE*-KO vs *APOE3* and *APOE4* vs *APOE3*. We observed significant overlap between expression patterns in the *APOE*-KO and *APOE2* comparison (Spearman's rank correlation, rho = 0.55, $p < 2.2e^{-16}$; Fig. 2d), as well as for the *APOE*-KO and *APOE4* variants (Spearman's rank correlation, rho = 0.34, $p < 2.2e^{-16}$; Fig. 2e). The strongest overlap was seen for genes downregulated in both *APOE2* and *APOE4*, suggesting that the mechanisms underlying the different APOE isoforms can be in part explained by a loss of APOE function. However, there were also unique transcriptional changes occurring in *APOE2* and *APOE4* microglia (Fig. 2d, e). This corroborates findings reported by Machlovi et al.[32] in which the authors performed similar analyses investigating the APOE4 and APOE3 isoforms in mouse microglia. To further explore the overlaps with the *APOE*-KO, we correlated the genes that had similar expression profiles in *APOE2* and the *APOE*-KO, and *APOE4* and the *APOE*-KO, when compared to the *APOE3* allele. All genes that had the same direction of expression change in *APOE2* and the *APOE*-KO, also had the same direction of expression change in *APOE4*, when compared with *APOE3* (Fig. 2f). A few genes were only significant in either *APOE2* or *APOE4*, including *TSPAN13*, which was upregulated in *APOE4* microglia and is associated with lipid accumulation in microglia[22,33]. Due to the pleiotropic nature of *APOE*, we evaluated whether genes differentially expressed across the *APOE* groups exhibited differential enrichment for AD genetic risk variants. Using MAGMA gene set analysis[34], we found that genes downregulated in the *APOE4* and *APOE*-KO microglia were enriched for risk variants identified in one AD GWAS[35] (FDR < 0.05; Fig. 2g, Supplementary Data 3). This supports previous evidence suggesting that the mechanisms of APOE4 reflect a loss-of-function in the context of AD[16,23,36], with APOE4 affecting biological pathways that are consistent with those linked to the polygenic component of AD.

To evaluate the upstream regulatory mechanisms associated with the transcriptomic changes across the APOE isoforms, we next investigated changes in chromatin accessibility. When compared to *APOE3*

microglia, *APOE2* microglia had 40 differentially accessible regions (DARs) (24 up, 16 down, FDR < 0.05; Fig. 3a, Supplementary Data 4), *APOE4* microglia had 50 DARs (38 up, 12 down, FDR < 0.05; Fig. 3b, Supplementary Data 4). Again, the direct comparison of *APOE4* with *APOE2* revealed the most differences, with 72 DARs (52 up, 20 down, FDR < 0.05; Supplementary Fig. 4a, Supplementary Data 4), with the fewest changes observed in the KO (14 up, 8 down, FDR < 0.05; Supplementary Fig. 4b). Notably, our analysis revealed consistent epigenomic and transcriptomic responses across microglia expressing the different APOE isoforms (Supplementary Fig. 5). For instance, *CHCHD2*, a mitochondrial gene involved in promoting cellular migration[37] and implicated in Parkinson's disease (PD)[38,39], was significantly downregulated in *APOE4* microglia when compared to both *APOE3* (logFC = −7.8, $p = 1.3e^{-11}$; Figs. 2b, 3c, Supplementary Data 2) and *APOE2* (logFC = −7.4, $p = 7.8e^{-11}$; Figs. 2c, 3c, Supplementary Data 2). In parallel, chromatin accessibility was significantly reduced close to the TSS of this gene when compared to *APOE3* (logFC = −7.8, $p = 1.2e^{-7}$, distance to TSS = 0; Fig. 3b, d, Supplementary Data 4) and *APOE2* (logFC = −6.5, $p = 8.3e^{-9}$, distance to TSS = 0; Supplementary Fig. 4a, Supplementary Data 4). Expression of this gene was also significantly reduced in the *APOE*-KO, suggesting a potential loss of protective function via this gene in the *APOE4* microglia (Supplementary Fig. 3). Similarly, the zinc finger protein *ZNF248* was upregulated in *APOE4* microglia (*APOE4* vs *APOE3*, logFC = 9, $p = 5.1e^{-14}$; *APOE4* vs *APOE2*, logFC = 7.9, $p = 5.2e^{-19}$; Supplementary Data 2) and genomic regions in the vicinity of this gene had increased chromatin accessibility (*APOE4* vs *APOE3*, logFC = 6.3, $p = 2.3e^{-5}$; *APOE4* vs *APOE2*, logFC = 6.6, $p = 3.2e^{-5}$; Supplementary Data 4). Interestingly, in an in vitro study investigating functional and transcriptional phenotypes of a *TREM2* mutant and knockout in iPSC-derived microglia-like cells, *ZNF248* was upregulated in the *TREM2*-KO, while *CHCHD2* expression was reduced in the R47H mutant[40]. For an overall assessment of the concordance between ATAC-seq and RNA-seq, we correlated the logFC values between DEGs and DARs at the corresponding promoter peaks. We observed a strong correlation across all *APOE* comparisons, indicating general concordance between changes in chromatin and gene expression (Supplementary Fig. 6). Microglia-specific regulatory regions originating from human samples are strongly enriched for AD genetic risk[5,6,10,41]. To evaluate whether human iPSC-derived microglia xenotransplanted into the mouse brain would recapitulate this enrichment we used stratified linkage disequilibrium score regression (s-LDSC)[42] with AD GWAS[43]. We found that open chromatin regions from the xenotransplanted microglia were enriched for AD heritability (FDR < 0.05, Fig. 3e, Supplementary Data 5). As the xenotransplanted microglia are predominantly responding to Aβ in our model, this enrichment also suggests that a significant proportion of AD risk is associated with microglial reactions to this pathological hallmark[44]. By repeating the analysis using GWAS data for autism spectrum disorder[45] and amytrophic lateral sclerosis (ALS)[46], we confirmed that this enrichment was specific to AD, and not a general brain disease enrichment (Fig. 3e, Supplementary Data 5).

## APOE2 and APOE4 microglia show differential expression of cytokines

Human microglia from the xenotransplantation model used here were previously profiled using single-cell RNA sequencing[27]. Mancuso et al. (2024) report eight microglial states responsive to Aβ pathology, including previously characterised disease-associated microglia (DAM), as well as novel states annotated as cytokine response (CRM) and antigen-presenting response (HLA) microglia. Using hypergeometric testing, we found that genes dysregulated across the microglia with different APOE isoforms were strongly enriched within several microglia clusters (Fig. 4a): The strongest association was observed for genes downregulated in *APOE4* microglia, which were enriched in the HLA, ribosomal microglia (RM), and DAM clusters (Fig. 4a).

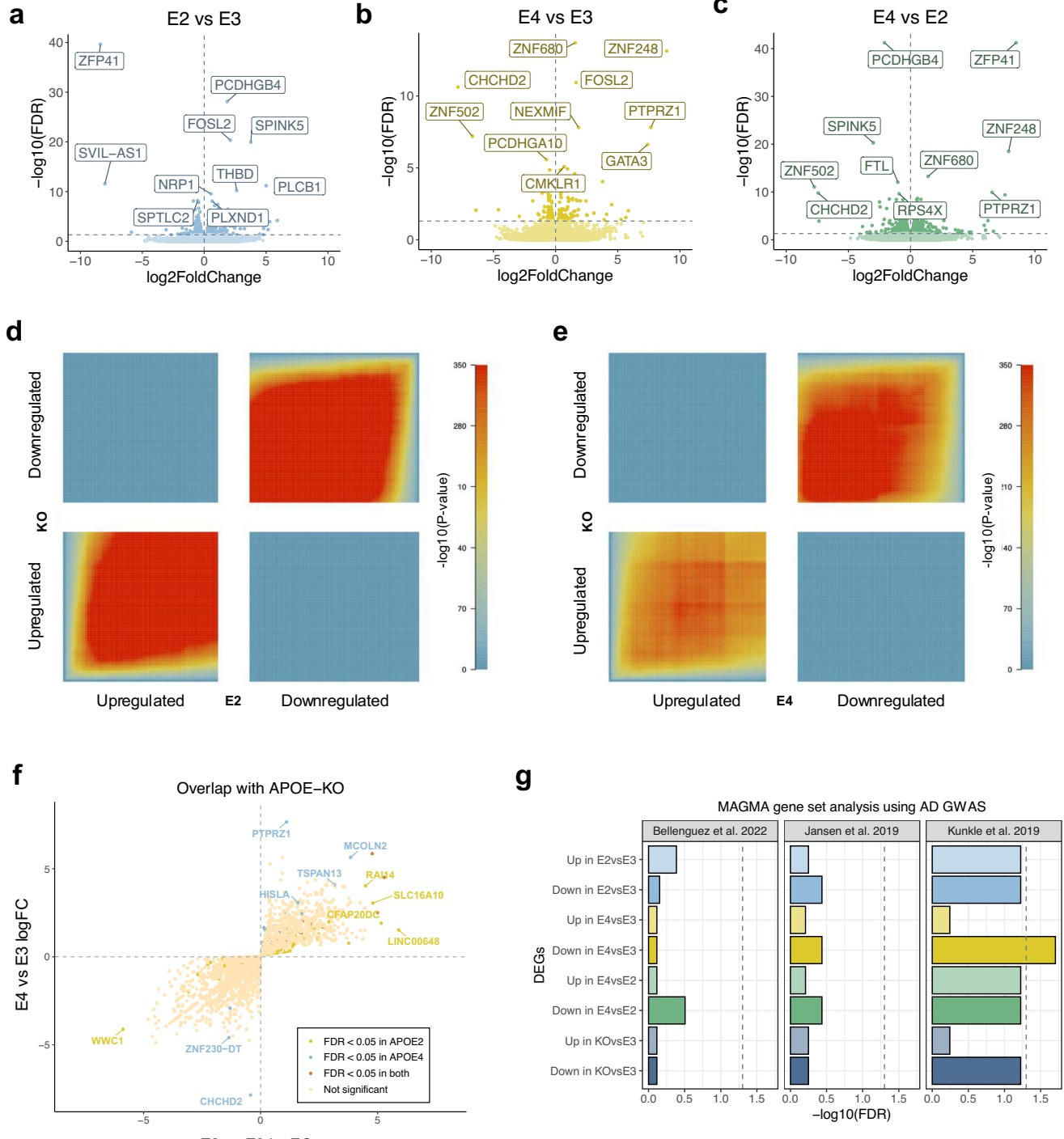

**Fig. 2 | Microglia exhibit widespread differences in gene regulation across the different APOE isoforms. a** Differentially expressed genes in *APOE2* vs *APOE3* microglia. **b** Differentially expressed genes in *APOE4* vs *APOE3* microglia. **c** Differentially expressed genes in *APOE4* vs *APOE2* microglia. **d** RRHO heatmap comparing expression signatures between *APOE*-KO vs *APOE3* and *APOE2* vs *APOE3*. **e** RRHO heatmap comparing expression signatures between *APOE*-KO vs *APOE3* and *APOE4* vs *APOE3*. The colour of the heatmap is reflective of the strength of the correlation based on *p*-value. **f** Scatterplot of *APOE4* vs *APOE3* logFC against *APOE2* vs *APOE3* logFC for genes with expression profiles overlapping with the *APOE*-KO. **g** Multi-marker Analysis of GenoMic Annotation (MAGMA) gene set analysis using the differentially expressed genes across the *APOE* alleles with three independent Alzheimer's disease (AD) genome-wide association study (GWAS)[35,43,87]. Source data are provided as a Source Data file. For the RRHO analysis, a one-sided hypergeometric test was used and raw *p*-values were plotted in the heatmap.

Furthermore, genes downregulated in *APOE4* microglia were enriched in the DAM cluster associated with negative regulation of tumour necrosis (TNF) cytokine production (Fig. 4b). Since negative regulation of cytokine production refers to processes that inhibit cytokine production, the downregulation of these genes in *APOE4* microglia

suggests increased cytokine production in this isoform, which was confirmed in the gene expression data (Fig. 4c–f). CRM microglia mount a pro-inflammatory response driven by the upregulation of chemokines and cytokines, and have only been characterised in human[16]. Mancuso and colleagues[16] showed that in response to Aβ,

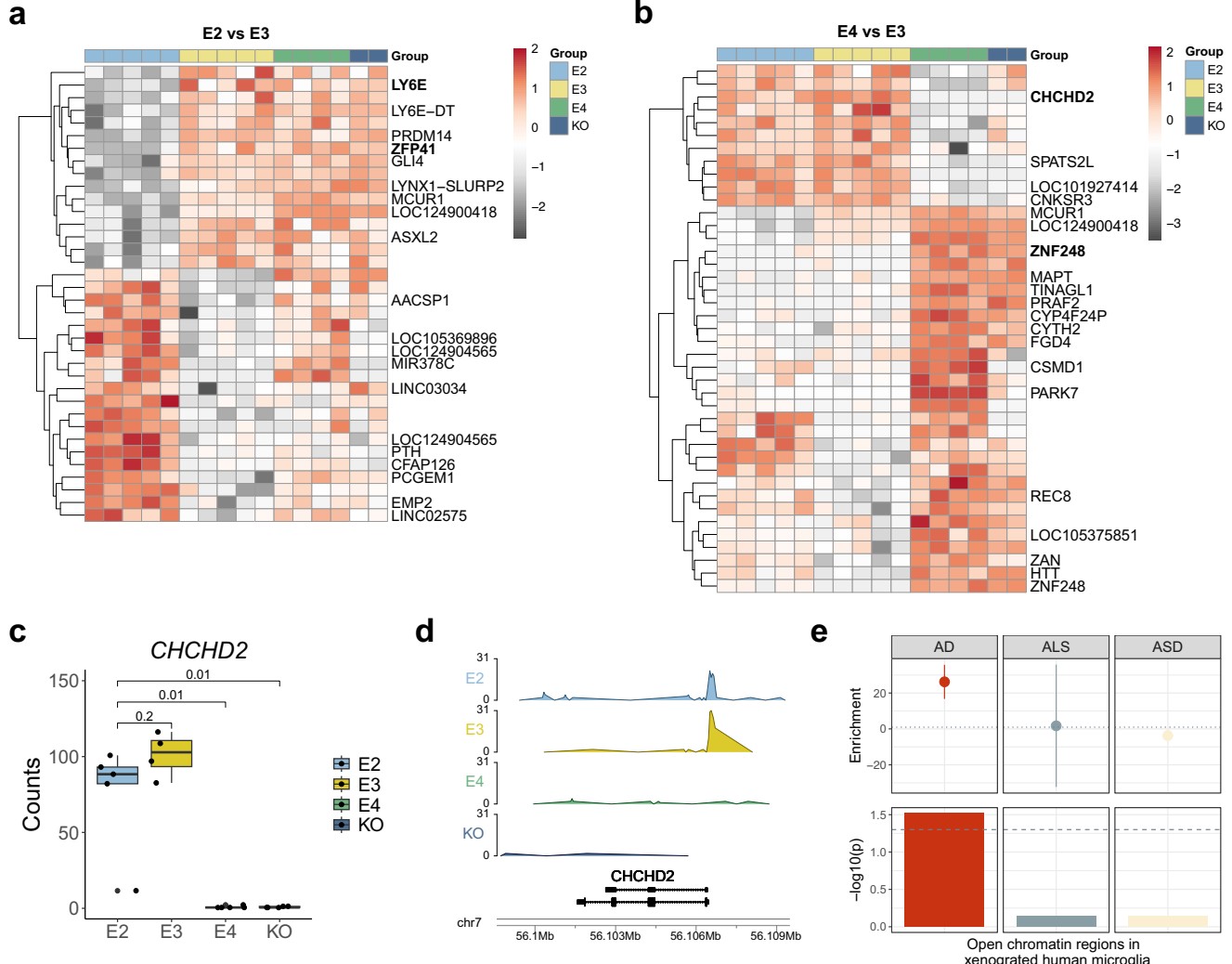

**Fig. 3 | Differentially expressed genes exhibit differential chromatin accessibility in their vicinity. a**, **b** Heatmaps showing differential chromatin accessibility of significant peaks (FDR < 0.05) when comparing (**a**) *APOE2* vs *APOE3*, and (**b**) *APOE4* vs *APOE3*. Shown are the genes annotated to the top 20 most significant peaks. Genes marked in bold were also significantly differentially expressed in the RNA-seq analysis. **c** Boxplot of expression profiles of *CHCHD2* shows reduced expression in the *APOE4* and the *APOE*-KO microglia. Expression profiles were derived from biological replicates: *APOE2* = 5, *APOE3* = 4, *APOE4* = 5, *APOE*-KO = 3. The two-sided Wilcoxon rank-sum test was used to calculate *p*-values. The central mark and edges indicate the 50th (median), 25th and 75th percentiles. Whiskers correspond to 1.5 * the IQR. **d** Genome tracks of chromatin accessibility signals around the *CHCHD2* locus show a loss of the open chromatin peak at the *CHCHD2* promoter in *APOE4* and KO. **e** S-LDSC analysis using all open chromatin regions from the xenotransplanted human microglia with genome-wide association study (GWAS) summary statistics for Alzheimer's disease (AD), amyotrophic lateral sclerosis (ALS), and autism spectrum disorder (ASD) shows a microglia-specific enrichment for AD. Data are presented as enrichment values +/− SD. Open chromatin regions were derived from biological replicates: *APOE2* = 5, *APOE3* = 4, *APOE4* = 5, *APOE*-KO = 2. Source data are provided as a Source Data file.

*APOE4* microglia shift to the CRM state rather than HLA. In agreement, Machlovi et al. (2022) report increased cytokine production in *APOE4* microglia. Conversely, where the authors found increased *TNFα* in *APOE4* microglia, TNF family members, including *TNFRSF25* and *TNFRSF21*, were upregulated in *APOE2* microglia in our study. Taken together with the decreased HLA and DAM response in *APOE4* microglia here, these findings suggest that *APOE4* microglia fail to transition towards microglial states that are thought to be protective[14,16] (Fig. 4a).

### *APOE2*-expressing microglia are associated with increased cellular migration and phagocytosis

We used weighted gene co-expression network analysis (WGCNA)[47] to identify microglial gene modules with similar expression profiles (Fig. 5a). These modules were then tested for differential expression across the *APOE* groups, and the differentially expressed modules were

functionally characterised using pathway enrichment analysis. We identified two differentially expressed modules significantly associated with GO biological processes. First, a gene module upregulated in *APOE2* microglia when compared to both *APOE3* and *APOE4* was associated with proliferation and cellular migration pathways (Fig. 5b, c, Supplementary Data 6). Such pathways are likely important for microglia being recruited towards the site of Aβ pathology and initiating its clearance, which in some cases, requires APOE[48]. Second, a module upregulated in *APOE2* when compared to *APOE4* was significantly associated with a range of immune responses, including both innate immune responses such as complement activation, and adaptive immune responses such as antibody-mediated immunity (Fig. 5d, e, Supplementary Data 6).

Previous studies have linked the APOE2 isoform to enhanced phagocytic capabilities[49–51]. Using a recently generated scRNA-seq dataset from phagocytic microglia associated with Aβ plaques[52], genes

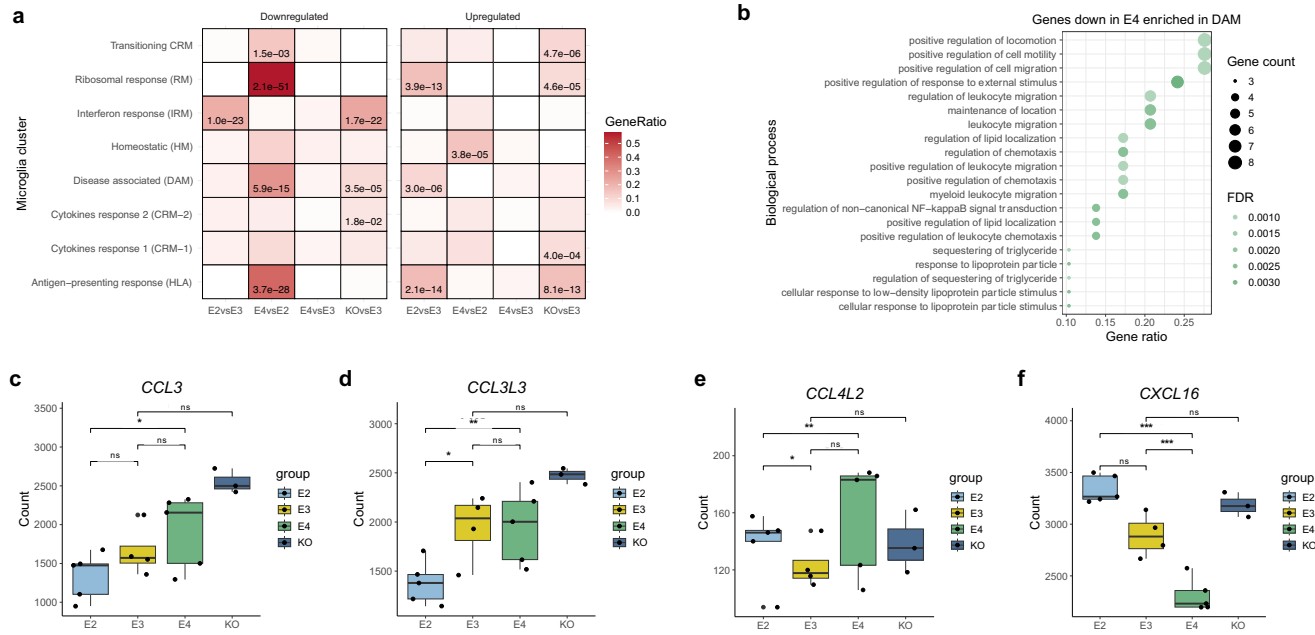

**Fig. 4 | Pro-inflammatory cytokines are upregulated in *APOE4* microglia.** **a** Heatmap showing enrichment of genes differentially expressed across the *APOE* groups amongst microglia clusters defined by scRNA-seq. **b** Dotplot of pathway enrichment analysis using genes downregulated in *APOE4* microglia that are enriched in the disease associated microglia (DAM) cluster. **c–f** Boxplots of gene expression profiles of cytokines (**c**) *CCL3*, (**d**) *CCL3L1*, (**e**) *CCL4L2*, and (**f**) *CXCL16*.

Expression profiles were derived from biological replicates: *APOE2* = 5, *APOE3* = 4, *APOE4* = 5, *APOE-KO* = 3. FDR corrected *p*-values were derived from the differential expression analysis using DESeq2: *p* < 0.05 (*), *p* < 0.01 (**), *p* < 0.001 (***), and for each comparison, can be found in Supplementary Data 2. In the boxplots the central mark and edges indicate the 50th (median), 25th and 75th percentiles. Whiskers correspond to 1.5 * the IQR. Source data are provided as a Source Data file.

upregulated in *APOE2*-expressing microglia when compared to both *APOE3* and *APOE4*, were exclusively overrepresented in a set of genes upregulated in phagocytic microglia (Fig. 6a). To functionally validate this finding, we performed phagocytosis assays using iPSC-derived human microglia using pHrodo E. coli particles and fluorescent myelin. As hypothesised, *APOE2*-expressing microglia internalised significantly higher amounts of pHrodo E. coli compared to all other *APOE* groups (Fig. 6b, c, Supplementary Fig. 7). Additionally, a significantly higher proportion of *APOE2*-expressing microglia successfully took up myelin compared to *APOE4*-expressing microglia (Fig. 6d, e), as reported previously[51]. We also observed that *APOE*-KO microglia displayed very high amounts of intracellular myelin. This could indicate an impairment to digest the phagocytosed material, and is in line with previous studies showing lipid accumulation in ApoE−/− mouse microglia in models of demyelination[53]. Overall, the enrichment of proliferation, migration, phagocytosis, and immune responses suggests enhanced microglial function in *APOE2*.

**Vitamin D receptor binding is upregulated in *APOE2* microglia**
To better understand the regulatory machinery of human microglia in Alzheimer's disease and the upstream orchestrators of altered transcriptional states, we performed de novo motif enrichment analysis using HOMER[54] on the open chromatin regions from the xenotransplanted microglia. Specifically, we used the top 100 peaks with increased and decreased chromatin accessibility across the APOE isoforms as input and defined all ATAC-seq peaks as the background set. Regions with increased accessibility in the *APOE2* microglia were strongly enriched for the DNA binding motif of the vitamin D receptor (VDR; Fig. 7a), a ligand-inducible transcription factor (TF) and main mediator of vitamin D signalling[55]. Importantly, vitamin D deficiency has been linked to increased risk for AD[56,57]. We next assessed whether VDR target genes were upregulated in *APOE2* microglia. A hypergeometric test confirmed an overrepresentation of genes upregulated in *APOE2* when compared to both *APOE3* and *APOE4* (Fig. 7b), in a list of

monocytic VDR target genes identified in a previous study[58]. This confirms the expected downstream transcriptional response predicted by increased VDR binding in *APOE2* microglia. Activation of an anti-inflammatory microglia phenotype via IL-10 and vitamin D signalling has been reported previously[59] (Fig. 7c). Consistent with this, in our data, the alpha subunit of the IL-10 receptor (*IL-10RA*) was significantly upregulated in *APOE2* microglia (Fig. 7d). We also checked to see whether VDR target genes were associated with any of the WGCNA modules identified in this study (Supplementary Fig. 8). The ME1 (dark green) module was enriched for upregulated VDR target genes, and this module was also identified as upregulated in *APOE2*-expressing microglia. The ME1 module was characterised by biological pathways associated with immune responses (Fig. 5d, e). The ME2 (blue) module was also enriched for upregulated VDR target genes, however, this module was not differentially expressed across the *APOE* groups (FDR < 0.05). Aside from the VDR enrichment, regions with increased chromatin accessibility in *APOE4*-expressing microglia when compared to *APOE2* were enriched for STAT2, a TF involved in interferon response signalling[60]. De novo motif enrichment analysis results for all APOE isoform comparisons can be found in Supplementary Data 7.

## Discussion
Increasing evidence points to a highly complex response of microglia to AD pathology. Here, we show that the human *APOE2*, *APOE3*, and *APOE4* differentially regulate microglia in the context of Aβ aggregates. By profiling human microglia isolated from a xenotransplantation model of AD using RNA-seq and ATAC-seq, we uncovered widespread changes to the transcriptomic and chromatin landscape of this cell type, dependent on the APOE isoform expressed. As anticipated, the largest differences were observed when comparing the AD risk opposing *APOE2* and *APOE4* microglia.

First, we observed consistent epigenomic and transcriptomic responses for several genes, including *CHCHD2* and *ZNF248*. *CHCHD2* is involved in promoting cell migration[37] and has been linked to familial

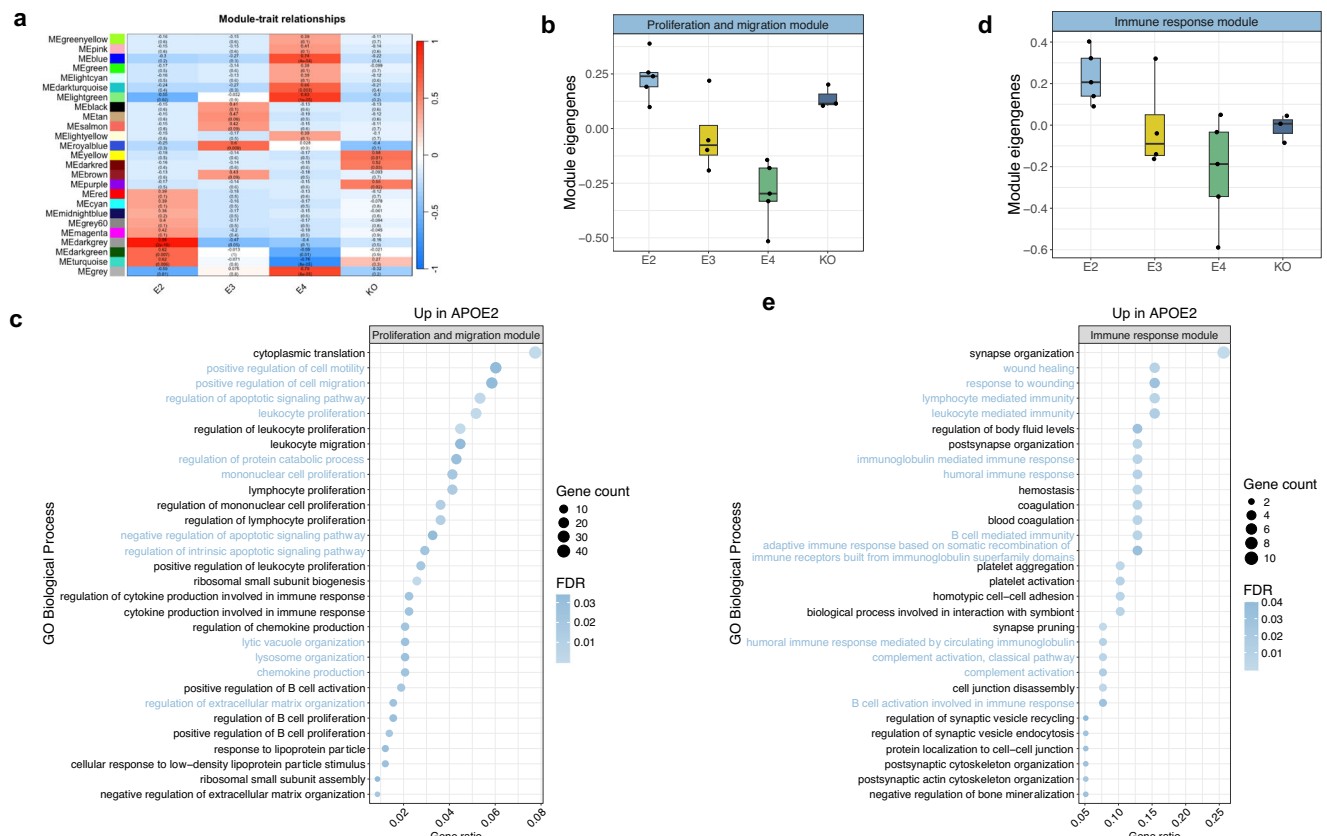

**Fig. 5 | Gene networks upregulated in *APOE2* microglia are associated with cellular migration and immune responses. a** WGCNA heatmap showing the correlation coefficients and *p*-values of modules across the *APOE* groups. The turquoise and the dark green modules correspond to the proliferation and migration module and the immune response module, respectively. **b** Boxplot of eigengene expression of the WGCNA module associated with proliferation and migration. The central mark and edges indicate the 50th (median), 25th and 75th percentiles. Whiskers correspond to 1.5 * the IQR. Expression profiles were derived from biological replicates: *APOE2* = 5, *APOE3* = 4, *APOE4* = 5, *APOE*-KO = 3). **c** Gene ontology (GO) biological processes enriched for genes within the WGCNA module

associated with proliferation and migration that is upregulated in *APOE2* microglia. **d** Boxplot of eigengene expression of the WGCNA module associated with immune responses. The central mark and edges indicate the 50th (median), 25th and 75th percentiles. Whiskers correspond to 1.5 * the IQR. Expression profiles were derived from biological replicates: *APOE2* = 5, *APOE3* = 4, *APOE4* = 5, *APOE*-KO = 3). **e** GO biological processes enriched for genes within the WGCNA module associated with immune responses, which was upregulated in *APOE2* microglia. In the heatmaps, Pearson correlation coefficients and Fisher's asymptotic *p*-values are shown. For the pathway enrichment analysis, *p*-values were computed using a one-sided Fisher's exact test. Source data are provided as a Source Data file.

and sporadic Parkinson's disease (PD)[38], where it is transcriptionally downregulated. Its decreased expression and chromatin accessibility in *APOE4* microglia, but also in the knockout, suggest a loss of normal function. Conversely, *ZNF248* was upregulated in *APOE4* and the knockout in both assays, suggesting a potential gain of toxic function. In a study comparing the effects of *TREM2* knockout and a *TREM2* mutation in a model of human microglia, the *TREM2* knockout had deficits in phagocytosis, chemotaxis, and survival that were not observed in the *TREM2* mutant[40]. *ZNF248* was one of only four differentially expressed genes with reduced expression in the knockout but increased expression in the mutant. Although the authors argue that it is unlikely that such a limited number of genes, including *ZNF248*, could explain such vast phenotypic differences[40], the overlap between *APOE4* microglia and *TREM2* knockout microglia is interesting. The convergence of results between the DEGs and DARs highlights the robustness of using multiple independent assays to profile cellular states in a disease context. Overlapping expression signatures with the *APOE* knockout enabled us to infer whether the *APOE2* and *APOE4* alleles resulted in a loss or gain-of-function. Although both alleles overlapped with the knockout, only genes downregulated in *APOE4* microglia and the knockout were both enriched for AD genetic risk, lending support to previous reports of *APOE4* microglia increasing AD risk through loss-of-function mechanisms[16,23]. Further investigation

into the genes shared between the knockout and *APOE4* microglia highlighted a strong upregulation of *TSPAN13*. This gene is also upregulated in microglia that accumulate damaging lipid droplets in the ageing brain[33], and in microglia homozygous for *APOE4*, in response to Aβ[22]. Our data suggest that lipid dysregulation in *APOE4* microglia may be driven by a loss of function.

Genes downregulated in *APOE4* were enriched within distinct microglial states identified in response to Aβ: HLA, RM, and DAM[27]. HLA represents a novel, human-specific microglial state that has a pronounced response to Aβ pathology and is thought to play a protective role[27]. RM are enriched for ribosomal genes. In murine AD models, stage 2 DAM cells, which signal the full activation of the DAM programme that is thought to be protective, are enriched for ribosomal genes[14]. When considered collectively, these enrichments suggest that *APOE4* microglia fail to shift into protective states. Furthermore, our analyses point toward diminished migratory capacity in *APOE4* and enhanced migratory capacity in *APOE2* microglia. This is supported in previous studies which have shown that migration is decreased in *APOE4* microglia-like cells[61], *APOE4* microglia have reduced motility and responsiveness to ATP, a chemotactic cue of neuronal damage, and amyloid beta[62]. In contrast, genes associated with microglial migration are upregulated in *APOE2* microglia[51]. Previously, *APOE4* microglia were shown to downregulate their

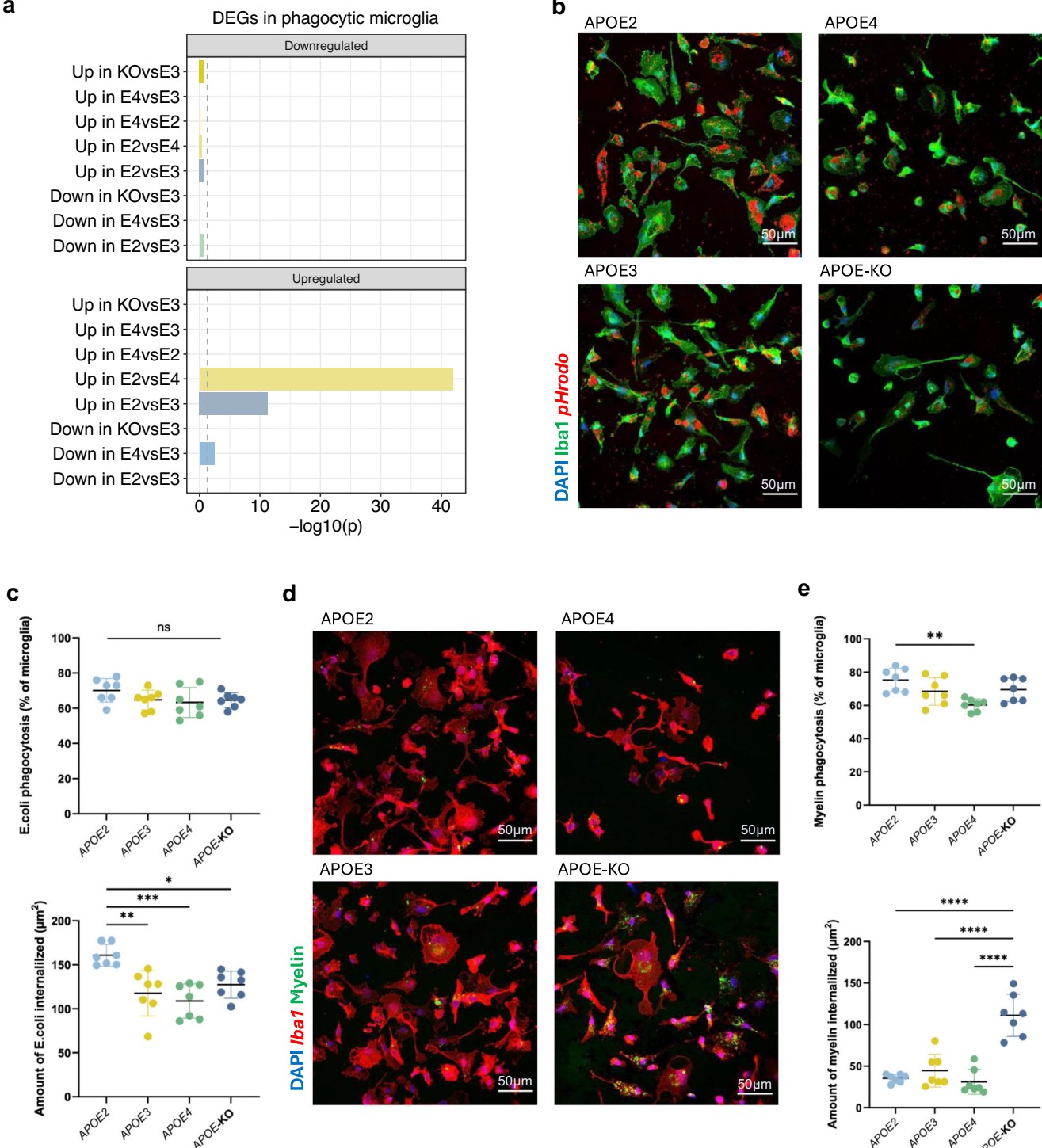

**Fig. 6 | iPSC microglia harbouring different *APOE* allelic variants display differences in the uptake of E. Coli pHrodo and fluorescent myelin particles.**
**a** Barplot showing the overrepresentation of genes upregulated in *APOE2* microglia in differentially expressed genes (DEGs) in phagocytic microglia responsive to Aβ plaques[52]. The dashed line represents the significance threshold after FDR correction ($p < 0.05$). **b**, **c** Representative images and quantification of pHrodo E. Coli particles (100 μg/ml) uptake by *APOE2*, *APOE3*, *APOE4* and *APOE*-KO iPSC-derived microglia. Each data point represents an independent well ($n = 7$). An average of 1236 +/− 87 cells were quantified per well. Top panel: $p > 0.99$ (KO vs E3), $p = 0.98$ (E4 vs E3), $p = 0.41$, $p = 0.98$ (E4 vs KO), $p = 0.39$ (E2 vs KO), $p = 0.22$ (E4 vs E2). Bottom panel: $p = 0.0014$ (E3 vs E2), $p = 0.0002$ (E4 vs E2), $p = 0.015$ (E2 vs KO),

$p = 0.82$ (E4 vs E3), $p = 0.77$ (KO vs E3), $p = 0.28$ (E4 vs KO). **d**, **e** Representative images and quantification of PKH67-labelled myelin (200 μg/ml) uptake by *APOE2*, *APOE3*, *APOE4*, and *APOE*-KO iPSC-derived microglia. Each data point represents an independent well ($n = 7$). An average of 211 ± 9 cells were quantified per well. Top panel: $p = 0.26$ (E2 vs E3), $p = 0.0019$ (E4 vs E2), $p = 0.39$ (E2 vs KO), $p = 0.14$ (E4 vs E4), $p = 0.99$ (E3 vs KO), $p = 0.081$ (KO vs E4). Bottom panel: $p = 0.77$ (E2 vs E3), $p = 0.97$ (E4 vs E2), $p < 0.0001$ (E2 vs KO), $p = 0.52$ (E4 vs E3), $p < 0.0001$ (E3 vs KO), $p < 0.0001$ (E4 vs KO). Statistical analysis was performed using a one-way ANOVA, with Bonferroni correction for multiple comparisons; * $p < 0.05$, ** $p < 0.02$ and *** $p < 0.001$. Data are presented as the mean +/- standard error of the mean. Source data are provided as a Source Data file.

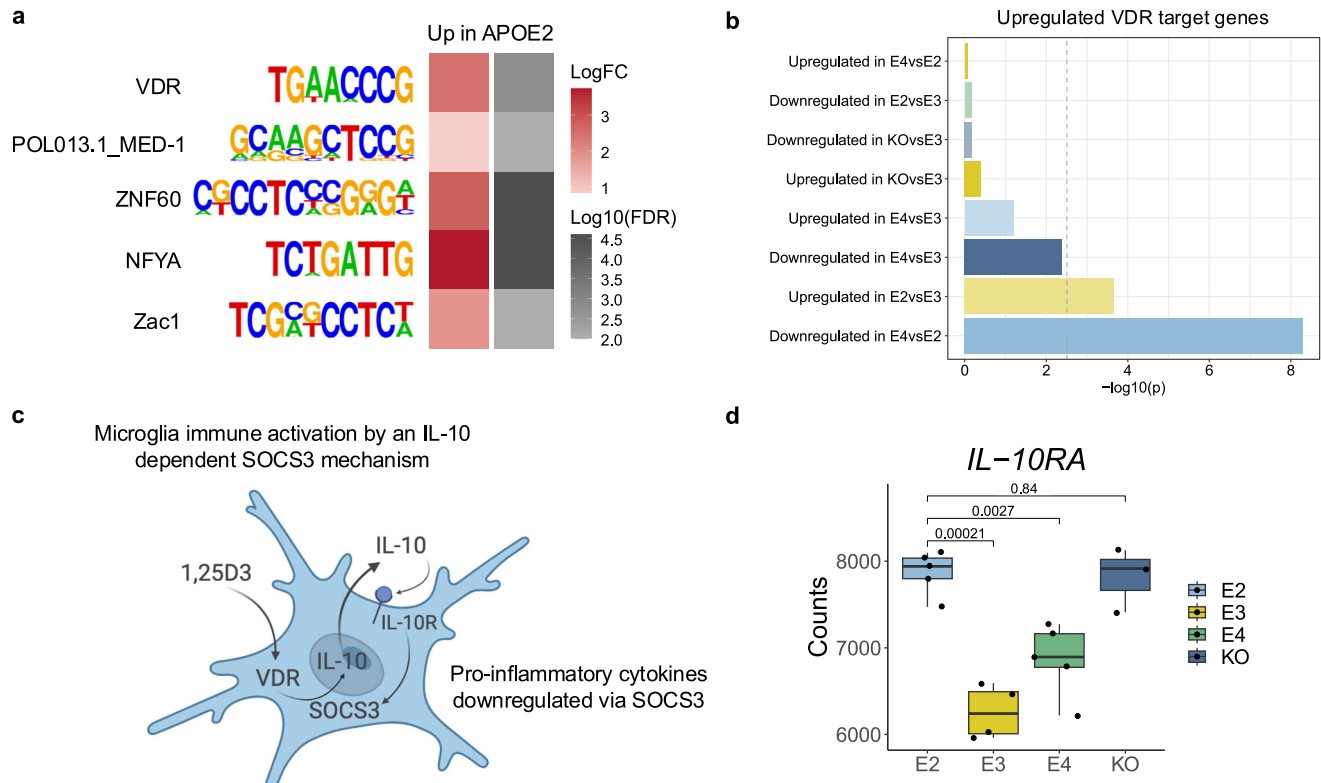

**Fig. 7 | Regions with increased chromatin accessibility in *APOE2* microglia are enriched for the binding of vitamin D receptor. a** Heatmap showing enrichment of motifs in regions with increased chromatin accessibility in *APOE2* microglia. **b** Barplot showing the overrepresentation of genes upregulated in *APOE2* microglia in a list of vitamin D receptor (VDR) target genes in monocytes[90]. The dashed line represents the significance threshold. P-values were computed using a one-sided hypergeometric test. **c** Illustration showing a mechanism of anti-inflammatory microglia activation mediated through vitamin D and IL-10 signalling[59]. Created in BioRender. Marzi, S. (2025) https://BioRender.com/x1lyqhw. **d** Boxplot of gene expression profiles of *IL-10RA*. The two-sided Wilcoxon rank-sum test was used to calculate *p*-values. Expression profiles were derived from biological replicates: *APOE2* = 5, *APOE3* = 4, *APOE4* = 5, *APOE*-KO = 3). The central mark and edges indicate the 50th (median), 25th and 75th percentiles. Whiskers correspond to 1.5 * the IQR. Source data are provided as a Source Data file.

expression of cellular migration genes in response to demyelination[51], and pericytes derived from *APOE4* carriers exhibited downregulation of genes associated with cellular migration[63]. This also suggests that migratory capacity is not a cell type or pathology-specific mechanism affected by APOE in AD. In concordance with signatures of cellular migration, *APOE2*-expressing microglia also exhibited enhanced phagocytic activity when compared to *APOE3* and *APOE4* microglia. This observation is in agreement with previous studies which report a lower accumulation of myelin debris in *APOE2*-TR mice[51], and increased clearance of Aβ from the interstitial fluid of mice expressing the APOE2 isoform[50] In addition, *APOE2*-expressing macrophages derived from transgenic mice with an *APP* mutation were also more efficient at degrading amyloid beta than both *APOE3* and *APOE4*-expressing macrophages[49].

Another mechanism through which *APOE4* may be exerting its pathogenic role in AD is by mounting a cytokine response. In other studies, mouse microglia expressing the humanised *APOE4* allele increased cytokine production[32], and xenotransplanted human microglia shifted towards a pro-inflammatory state[27]. Similarly, we report a general upregulation of cytokines in *APOE4* when compared to both *APOE2* and *APOE3* microglia. One exception was *CXCL16*, which was upregulated in the *APOE2* microglia. However, this chemokine has been reported to drive microglia to an anti-inflammatory phenotype in brain tumours[64]. In addition to changes to cytokine expression, regions with increased chromatin accessibility in *APOE4*-expressing microglia were enriched for STAT2, a TF involved in interferon response signalling. Increased interferon signalling as well as upregulation of associated genes has already been shown in mouse microglia

expressing *APOE4*[32]. Machlovi et al. (2022) also reported increased *TNFα* expression in *APOE4* microglia, while we observed the opposite— nine TNF family genes were increased in *APOE2* when compared to *APOE4* (Supplementary Data 1). Whether a cytokine is pro-inflammatory is context dependent, and it is therefore difficult to conclude whether the upregulation of these TNF family genes is driving a pro-inflammatory response. In addition, it remains unclear to what extent pro-inflammatory responses are protective versus pathogenic in AD, and what triggers the increased production of pro-inflammatory cytokines in the APOE isoforms. For instance, in *APOE2* microglia the upregulation of TNF family genes may be triggered by their migration towards and interaction with Aβ plaques. Whereas in *APOE4* microglia, the increased production of pro-inflammatory cytokines could be due to a lack of response to Aβ, which in turn would result in continued Aβ deposition and a sustained pro-inflammatory response.

Several studies have shown an enrichment of AD genetic risk within microglia-specific genes and regulatory regions from the human brain[5,6,11]. Here, using the chromatin accessibility profiles from the xenotransplanted microglia we recapitulate this enrichment, highlighting the robustness of this model for investigating human genetic risk in a disease context. At the level of the transcriptome, AD genetic risk was enriched within genes downregulated in *APOE4* microglia but also the *APOE* knockout. Supporting previous studies, this overlap suggests that AD risk increased by the presence of *APOE4* is partially mediated through loss of protective function. Our findings underscore the need to consider the interplay between genetic risk factors and microglial states in AD.

In addition to arguing for increased proliferation, migration, phagocytosis and immune response in *APOE2* microglia as underlying this isoform's protective effect, we report a potential upstream regulatory role for the VDR. In the context of AD, low levels of vitamin D have been associated with a higher incidence of the disease[56,57], and vitamin D supplementation has been shown to improve disease outcomes[65,66]. It's important to take into consideration *APOE* genotype, as some studies have shown that *APOE4* carriers have higher vitamin D levels[67,68] and therefore vitamin D supplementation may be more beneficial to non-carriers[66]. Vitamin D acts via binding to VDR, and enrichment of VDR in regions with increased chromatin accessibility in *APOE2* may therefore enable these microglia to be more responsive to vitamin D, regardless of serum levels. Furthermore, the increased expression of the *IL-10RA* in *APOE2* microglia was particularly interesting. Vitamin D, via the VDR, increases the expression of the anti-inflammatory cytokine *IL-10*[59]. IL-10 then activates SOCS3 via the IL-10 receptor, and this mechanism suppresses the expression of pro-inflammatory cytokines. Several other studies have also demonstrated an association between vitamin D and the expression of anti-inflammatory factors in microglia[69–74]. The functional role of VDR activation and binding warrants further studies in terms of mechanisms and therapeutic exploration.

Our study has a number of limitations. First, microglia exist in different subtypes and states. For example, microglia associated with Aβ plaques may have distinct transcriptomic and epigenomic profiles compared to less responsive microglia. While it is worth considering the contributions of *APOE* from mouse microglia and astrocytes, these should remain consistent across the *APOE* groups and, as such, should not affect differential expression and chromatin accessibility analyses. Although transcriptomic and epigenomic profiling provide valuable insights into gene regulatory mechanisms, other factors, such as histone modifications, also play a significant role in AD[3,4,41]. It is also worth considering that the APOE isoforms might exert risk or protection prior to the onset of Aβ pathogenesis. Future studies could adopt a time-series approach to investigate this. We linked regions with differential chromatin accessibility to differentially expressed genes based on their proximity. While this approach may capture promoter-gene relationships, many DARs may also function as enhancers. These enhancer-gene links can regulate target genes up to a megabase away, making them more challenging to identify. Using appropriate chromatin interaction data such as Hi-C could help disentangle these connections. Finally, while the absence of an adaptive immune system is necessary to prevent xenograft rejection, it may lead to unaccounted for changes in the microglial response[75].

Our work sheds light on the regulation of microglia in AD: we show that it is dependent on APOE isoform, at both the level of the transcriptome and epigenome, further highlighting the complexity of this cell type in response to Aβ. Our work suggests that *APOE4* microglia have compromised microglial functions including diminished migratory capacity and heightened pro-inflammatory responses compared to *APOE2*, and these may underlie the increased risk of AD seen in carriers of this isoform. Furthermore, our findings underscore the importance of considering the interplay between genetic risk factors, such as *APOE*, and microglial states in disease progression. Importantly, we highlight the potential involvement of the VDR in modulating microglial responses, providing new avenues for therapeutic exploration. Overall, the use of the microglia xenotransplantation model coupled with genome-wide profiling has enabled us to dissect the regulatory landscape of microglia expressing the different APOE isoforms. In future, this approach could be extended to other relevant genes. In summary, our study emphasises the complex interplay between genetic, epigenetic, and environmental factors in shaping microglial responses in AD and underscores the need for targeted interventions based on *APOE* genotype.

## Methods

### Differentiation of microglial progenitors
BIONi010-C-2 (APOE e3/KO), BIONi010-C-3 (APOE KO/KO), BIONi010-C-4 (APOE e4/KO), BIONi010-C-6 (APOE e2/KO) were differentiated into microglial precursors using the MIGRATE protocol, described in detail in Fattorelli et al. (2021)[26]. The lines were obtained via Bioneer from the European Bank for induced pluripotent Stem Cells (EBiSC), where all originating tissue donors give fully informed consent for donation, generation of iPSCs and sharing of lines for research. This set of APOE isogenic cell lines is based on the male iPSC cell line BIONi010-C. Stem cells were cultured on Matrigel in E8 Flex medium, dissociated at ~70–80% confluence, and aggregated in U-bottom 96-well plates with BMP4, VEGF, and SCF. Embryoid bodies were then transferred to six-well plates and cultured in X-VIVO medium with additional cytokines, followed by a media switch on day 11. By day 18, microglial precursors were harvested and transplanted into P4 mouse brains after depleting endogenous microglia with CSF1R inhibitor BLZ945. Ethics approval for transplantation of human iPS cells was obtained by the Ethics Committee Research UZ / KU Leuven (study S65730).

### Human microglia xenotransplantation model
*App*[NL-G-F] mice were crossed with homozygous Rag2tm1.1Flv Csf1tm1(CSF1)Flv Il2rgtm1.1Flv Apptm3.1Tcs mice (Jacksons Lab, strain 017708) to generate the Rag2-/- Il2rγ-/- hCSF1KI *App*[NL-G-F] used in this study. The strain was maintained on the original C57Bl/6xBalbC background. In total, we transplanted 500,000 cells bilaterally across 20 mice. Male mice were used for the xenotransplantations to limit biological variability. Mice had access to food and water ad libitum and were housed with a 14/10 h light-dark cycle at 21 °C in groups of two to five animals. Animal experiments were approved by the local Ethical Committee of Laboratory Animals of the KU Leuven (government licence LA1210579 ECD project number P177/2017) following local and EU guidelines. Five biological replicates were prepared per experimental group: *APOE2*, *APOE3*, *APOE4*, *APOE* KO. Animals were anaesthetised with an overdose of sodium pentobarbital and transcardially perfused with ice cold PBS. Samples were obtained in the morning between 9 and 11 am. From each sample at 12 months, FACS purification of the following cell numbers were attained: 100,000 cells for ATAC-seq, 200,000 cells for RNA-seq. The FACS plots with the gating strategy can be seen in Supplementary Fig. 9. ATAC-seq samples were processed immediately after cell collection for tagmentation and elution of transposed DNA (details in ATAC-seq methods section). RNA-seq was conducted from cell pellets snap frozen in liquid nitrogen.

### RNA-seq library preparation
From snap frozen cell pellets of 200,000 cells per sample, RNA was extracted using the Monarch Total RNA Miniprep Kit (T2010) following the manufacturer's instructions. RNA-seq was conducted using the rRNA depletion strategy rather than mRNA enrichment so that non-coding RNAs could be recovered[76]. rRNA depletion was performed using NEBNext rRNA Depletion Kit v2 Human/Mouse/Rat with RNA Sample Purification Beads (E7405), followed by stranded (directional) library preparation using the NEBNext Ultra II Directional RNA Library Prep Kit for Illumina (E7765) following manufacturer's protocols without adjustments. RNA quality was checked using the Agilent RNA 6000 Pico Kit (5067-1513) and final libraries were assessed using Agilent High Sensitivity DNA Kit (5067-2646) where all libraries were appropriate for sequencing apart from one replicate of the *APOE3* microglia—therefore this isoform only has four biological replicates for RNA-seq.

### ATAC-seq library preparation
ATAC-seq was conducted as previously described[77]. Following FACS collection of 100,000 cells per sample, cells were spun down at 500 *g* for 5 min at 4 °C, and the supernatant was removed. Cell pellets were

gently resuspended in 50 μL of ice cold Lysis Buffer (10 mM Tris-HCl pH 7.4, 10 mM NaCl, 3 mM MgCl2, 0.1% IGEPAL CA-630). 2.5 μL Tagment DNA Enzyme (Illumina; 20034197) was added directly and gently mixed by pipetting. The transposition reaction was incubated at 37 °C for 30 min, then transferred to ice. DNA was purified immediately with the Zymo ChIP DNA Clean and Concentration Kit (D5205) following manufacturer's instructions. The DNA column was spun dry prior to elution of transposed DNA, which was conducted with 11 μL Elution Buffer. Purified DNA was stored at -20 °C until library preparation. 10 μL DNA per sample was transferred into a PCR tube and 34.25 μL PCR master mix was added per sample. 6.25 μL of 10 μM Nextera Primer 2 (with barcode) was added per sample, where a different barcode was used for each sample to enable multiplexing. PCR was conducted using the following settings: (1) 72 °C for 5 min, (2) 98 °C for 30 s; (3) 98 °C for 10 s; (4) 63 °C for 30 s; (5) 72 °C for 1 min; (6) repeat steps (3)-(5) for a total of 10 cycles; (7) hold at 4 °C. Amplified library was purified using the Zymo ChIP DNA Clean and Concentration Kit (D5205). The purified library was eluted using 20 μL Elution Buffer. 5 μL of 5x TBE Loading Buffer (Invitrogen; LC6678) was added and loaded in a 12-well 10% TBE gel (Invitrogen; EC62752BOX). A ladder was prepared using 0.25–0.5 μL ORangeRuler 50 bp DNA Ladder (ThermoFisher; SM0613) diluted in 5 μL 5x TBE loading buffer. The gel was run at 70 V until DNA enters the gel, then increased to 140 V for approximately one hour. The gel was stained using 10 mL 1x TBE with SYBR Gold Nucleic Acid Gel stain (Invitrogen; S11494) diluted at 1:10,000 (1 μL). The gel was cut between 175 and 225 bp markers into a 0.5 mL DNA LoBind tube perforated three times with a 22 G needle. The gel was shredded by centrifugation at maximum speed for 2 min at room temperature into a 1.5 mL DNA LoBind tube. 150 μL Diffusion Buffer (0.5 M Ammonium Acetate, 0.1% SDS, 1 mM EDTA, 10 mM Magnesium Acetate, ddH2O) was added to the gel in the 1.5 mL tube and shaken at room temperature for 45 min. The sample was then transferred to filter columns using wide-bore tips and spun at max speed for 2 min. DNA was purified (-140 μL) using the Zymo ChIP DNA Clean and Concentration Kit and eluted with 10 μL Elution Buffer into 1.5 mL DNA LoBind tubes. Final libraries were quantified with the Qubit 1X dsDNA HS Assay Kit (ThermoFisher; Q33230) and stored at −20 °C prior to sequencing (yield: ~0.25 ng/μL).

## Sequencing

Final library size distributions were assessed by Agilent 2100 Bioanalyser and Agilent 4200 TapeStation for quality control before sequencing. Libraries were pooled to achieve an equal representation of the desired final library size range (equimolar pooling based on Bioanalyser/TapeSation signal in the 150–800 bp range). Paired-end Illumina sequencing using the HiSeq 4000 PE75 strategy was conducted on barcoded libraries at the Imperial Biomedical Research Centre (BRC) Genomics Facility following the manufacturer's protocols.

## ATAC-seq QC and processing

General QC of each sample was assessed using fastQC (https://www.bioinformatics.babraham.ac.uk/projects/fastqc/), followed by adaptor trimming using TrimGalore! (https://github.com/FelixKrueger/TrimGalore). Reads were aligned to GRCh38 using bowtie2 with the following arguments: –local –very-sensitive –no-mixed –no-discordant -I 25 -X 1000. Post-alignment QC included removing: reads mapping to the mitochondrial genome, duplicate reads, multi-mapping reads, and reads with low mapping quality (q < 30). Read count generation was performed using featureCounts[78]. Additionally, peaks were filtered using the filterByExpr() function in DESeq2[31], retaining only peaks with sufficiently high counts for statistical analysis. This left 167,951 peaks for downstream analyses. The 2 APOE-KO samples with high APOE expression in the RNA-seq data were also excluded from the ATAC-seq

dataset. Additionally, another APOE-KO and one APOE4 microglia sample were discarded due to having a low number of reads after QC filtering (<9 million).

## RNA-seq QC and processing

General QC of each sample was assessed using fastQC (https://www.bioinformaticsf.babraham.ac.uk/projects/fastqc/), followed by adaptor trimming using TrimGalore! (https://github.com/FelixKrueger/TrimGalore). Reads were aligned to the GRCh38 genome and transcriptome using STAR[79]. Duplicate and multi-mapping reads were retained. Transcript quantification was performed using Salmon[80], using the GC bias flag. Two of the APOE knockout samples with high APOE expression were excluded from subsequent analyses (Supplementary Fig. 2).

## Differential expression and accessibility analysis

DESeq2[31] was used for the differential expression and differential chromatin accessibility analysis. DESeq2[31] was designed for the differential analysis of RNA-seq data and has since been widely used for this purpose. In a recent study comparing methods for differential analysis of ATAC-seq read counts, Gontarz et al (2020)[81] showed that with five replicates, which we have for most of our samples, DESeq2 had the lowest false positive rate and a true positive recall comparable to other methods available for differential accessibility analysis. For both analyses, the APOE3 microglia samples were used as a baseline for comparison, and we additionally tested for differences between APOE4 and APOE2 microglia. To perform the differential analysis, we used the DESeq() function which provides a wrapper for three functions: estimateSizeFactors() for estimation of size factors, estimateDispersions() for estimation of dispersion, and nbinomWaldTest() for negative binomial GLM fitting and Wald statistics. Genes and peaks were defined as being significant if $p < 0.05$ after FDR correction.

## Weighted gene co-expression network analysis

To identify which genes had similar expression profiles across the APOE groups, we used weighted gene co-expression network analysis (WGCNA)[47]. First, transcripts with zero or low expression counts were filtered out using the filterByExpr() function in edgeR[82]. As suggested by the authors of WGCNA, the count data was normalised by variance stabilising transformation and explored for outliers using principal component analysis. An appropriate soft thresholding power was chosen to ensure a scale-free network and used as input to the blockwiseModules() function in WGCNA to calculate the adjacency matrix. This function was also used to detect gene co-expression modules and to calculate module eigengenes. As defined by the authors[47], the module eigengenes are the first principal component of a given module and can be considered to represent the gene expression profile of that module.

## Functional enrichment analysis using differentially expressed WGCNA modules

The module eigengenes identified in the WGCNA analysis were used to perform differential expression analysis using the lmFit() function in limma[83], across the APOE groups. As the only differentially expressed modules were associated with APOE2, the genes belonging to these modules were used to perform pathway enrichment analysis using clusterProfiler[84], allowing characterisation of the APOE2-associated modules based on their gene ontology (GO) enrichments. GO terms were considered to be significantly associated with the given modules if $p < 0.05$ after FDR correction.

## Stratified linkage disequilibrium score regression

To estimate the proportion of disease SNP-heritability attributable to open chromatin regions in the xenotransplanted microglia, we

performed stratified linkage disequilibrium score regression (s-LDSC). Annotation files were generated and used to compute LD scores. Publicly available GWAS summary statistics for a recent AD GWAS[43] were downloaded and converted to the required format for LDSC. Steps for the analysis were followed as instructed here https://github.com/bulik/ldsc/wiki and required files were downloaded from https://alkesgroup.broadinstitute.org/LDSCORE/GRCh38. LDSC was run using the full baseline model, thereby computing the proportion of SNP-heritability associated with the annotation of interest, while taking into account all the annotations in the baseline model. As we observed a significant enrichment, we repeated the analysis using GWAS data for PD[85], Amyotropic Lateral Sclerosis[46], and Autism spectrum disorder[86], to ensure this was not a generic neurological enrichment.

## MAGMA gene set analysis

MAGMA gene set analysis[34] was used to assess the enrichment of AD SNP-based heritability among differentially expressed genes across the APOE isoforms. The SNP window was restricted to the gene region (0,0). Summary statistics for three independent AD GWAS were downloaded[35,43,87] and formatted for use with MAGMA using MungeSumstats[88]. *P*-values were corrected using the FDR method.

## In vitro phagocytosis assay

Microglia progenitors, as generated according to the MIGRATE protocol, were collected on day 18 and plated in differentiation medium (DMEM/F12 supplemented with 200 mM L-glutamine, 1:2000 human insulin, 5 ng/mL N-Acetyl-L-cysteine, 50 mg/mL apo-transferrin, 20 µg/mL sodium selenite, 1 µg/mL heparin sulphate, 50ng/mL M-CSF, 100 ng/mL IL-34, 2 ng/mL transforming growth factor (TGF)-β, 10 ng/mL fractalkine (CX3CL1), 1.5 ng/mL cholesterol) in a concentration of 3.75*105 cells/mL in 8 well chamber slides (Ibidi, 80841). On day 6, microglia were incubated with 100 µg/mL pHrodo Red E.coli Bio-Particles (Thermofisher, P35361) for 1 h, or 200 µg/mL human myelin debris, as isolated from human brain tissue according to previously published protocol[89] and fluorescently labelled according to instructions of the MINI67 cell membrane labelling kit (Sigma-Aldrich, MINI67) for 3 h. Cultures were rinsed with PBS three times and fixed in 4% paraformaldehyde for 15 min.

Fixed cells were permeabilized with 1% Triton X-100 in PBS for 5 min and nonspecific binding was blocked with 5% bovine serum albumin (BSA) in PBS for 10 min. Microglia were incubated with anti-ionized calcium-binding adaptor molecule 1 (Iba1; WAKO, 019-1974, 1:500), followed by incubation with secondary Alexa Fluor 488 or 594 donkey anti-rabbit antibodies (Invitrogen A21206 and A21207, 1:400), both in 1% BSA in PBS for 1 h at RT. This was followed by counterstaining with 1.67 µg/mL 4′,6-diamidino-2-phenylindole (DAPI; Merck, D9542) for 15 min at RT. Finally, cells were mounted with Mowiol solution and stored at 4 °C until imaging.

For each well (7 per genotype and phagocytosis stimulus), three fields were imaged using a Nikon A1R Eclipse Ti confocal microscope. For each field, a z-stack of three images was acquired with steps of 2.95 µm at 10x magnification for the E.coli conditions and 0.88 µm at 20x magnification for the myelin conditions. Images were further analyzed with a custom made FIJI script (v1.53). Maximum intensity projections were generated from z-stacks, and microglia masks were generated using the 'Analyze Particles' function on the Iba1 channel, followed by manual verification. The number of microglia positive for E. coli/myelin phagocytosis was counted with the use of the 'Cell Counter' plugin. Thereafter, the area of E. coli/myelin signal within microglia positive for phagocytosis was measured in µm². Values from three fields per well were averaged for analysis. Differences in phagocytosis between genotypes were assessed with one-way ANOVA and Bonferroni's multiple comparisons post-hoc test in GraphPad Prism (v10.2.0).

## Reporting summary

Further information on research design is available in the Nature Portfolio Reporting Summary linked to this article.

## Data availability

FASTQ and read count files have been deposited in the Gene Expression Omnibus (GEO) database under accession code GSE271384 for the ATAC-seq dataset and GSE271385 for the RNA-seq dataset. All supplementary data files are available on Zenodo: https://doi.org/10.5281/zenodo.15202660. Source data are provided with this paper. https://zenodo.org/records/15202660 Source data are provided with this paper.

## Code availability

All the data and code required to reproduce the figures in this manuscript are available in our GitHub repository: https://github.com/Marzi-Lab/APOE_microglia and on Zenodo: https://doi.org/10.5281/zenodo.15176132.

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

## Acknowledgements

This work was funded by an Alzheimer's Association grant (grant number ADSF-21-829660-C) to SJM and BDS. This work was also supported by the UK Dementia Research Institute award number UKDRI-6205 through UK DRI Ltd, principally funded by the UK Medical Research Council. SJM received funding from the Edmond and Lily Safra Early Career Fellowship Program (https://www.edmondjsafra.org) and the Medical Research Council (grant number MR/W004984/1). BDS has funding from a Medical Research Council Grant (MR/Y014847/1), Fonds Wetenschappelijk Onderzoek (FWO) #12P5922N, Methusalem grant 3M140280, European Research Council (ERC) under the European Union's Horizon 2020 research and innovation programme (grant agreement No. 834682 CELLPHASE_AD). RM has funding from the European Research Council (ERC) under the European Union's Horizon 2020 Research and Innovation Programme (project no. 101041867—XenoMicrogliaAD), Fonds voor Wetenschappelijk Onderzoek (grants no. G0C9219N, G056022N and G0K9422N) and is a recipient of a postdoctoral fellowship from the Alzheimer's Association USA (fellowship no. 2018-AARF-591110 and 2018-AARF-591110-RAPID). He also receives funding from the Alzheimer's Association (E2A-23-1148152, 23AARF-1026404 and ABA-22-968700), BrightFocus Foundation (A2021034S), SAO-FRA (grant no. 2021/0021) and the University of Antwerp (BOF-TOP 2022-2025). KBM is a recipient of an MRC Doctoral Training Partnership award. We thank Annerieke Sierksma for helpful discussions and feedback. We thank the members of the International Neuroimmune Consortium for their insightful conversations and support in this project. The Imperial BRC Genomics Facility has provided resources and support that have contributed to the research results reported within this paper. The Imperial BRC Genomics Facility is supported by NIHR funding to the Imperial Biomedical Research Centre.

## Author contributions

S.J.M., B.D.S. and R.M. designed the study. D.H., L.W., S.R., G.L.F., and R.M. performed experiments. K.B.M. analyzed the results. I.G. contributed to the study design and data interpretation. S.J.M., B.D.S., R.M. supervised the project. K.B.M. and S.J.M. wrote the manuscript with

input from all coauthors. All authors reviewed and approved the final manuscript.

## Competing interests

B.D.S. is or has been a consultant for Eli Lilly, Biogen, Janssen Pharmaceutica, Eisai, AbbVie and other companies. B.D.S. is also a scientific founder of Augustine Therapeutics and a scientific founder and stockholder of Muna Therapeutics. The remaining authors declare no competing interests.
