## [Transparent Peer Review file · Nature Communications]

The APOE isoforms differentially shape the transcriptomic and epigenomic landscapes of human microglia xenografted into a mouse model of Alzheimer's disease

Corresponding Author: Dr Sarah Marzi

Version 0:

Reviewer comments:

Reviewer #1

(Remarks to the Author)

This study by Murphy et al. utilizes the xenotransplantation model developed by the de Strooper lab to study epigenetic regulation of human microglial phenotypes driven by different APOE isoforms in the context of Abeta pathology, reporting some interesting findings with regards to cellular response regulation in APOE2 and APOE4 vs. APOE3 cells. Through comparison with APOE ko cells, their study also allows for interpretation with regard to whether cellular phenotypes are driven by loss of function. While the paper certainly adds novel insights into how APOE isoforms cell-autonomously affect microglial responses, some concerns need to be addressed before publication.

Major concerns:

1) The authors report exclusion of two APOE ko samples that showed no reduction in APOE gene expression. While I applaud the authors for their honesty, this raises concerns regarding the identity of the transplanted iPS-derived microglia. This is particularly the case, because the APOE4 group also shows two samples with conspicuously higher APOE expression levels as well as two samples that cluster closer to APOE ko and APOE2 cells in the PCA and hierarchical clustering analyses. The genotype of all samples needs to be verified (using the existing sequencing data or qPCR) to ensure that these cells are indeed the correct genotype. Otherwise, the data cannot be interpreted.

Along the same line of thought, Figure 1c shows some ATAC reads within the gene body of the APOE gene in APOE ko samples and very clearly in the promoter region. Is the promoter region still present in this knockout line? Please provide more details.

2) No supplementary tables were submitted, preventing assessment of some of the results.

3) All experiments are performed in APP transgenic animals, so it is not possible to conclude which molecular alterations are due to the different APOE isoforms per se or due to the Abeta pathology x APOE isoform interaction. This needs to be made clear throughout the text.

4) The authors report only few DARs compared to the number of DEGs. How was the fold-change cut-off chosen for DAR detection? Under conditions of chronic inflammation and because the samples were analyzed in bulk with only a subset of cells responding to Abeta pathology, one might expect a lower FC than found in equivalent experiments during acute and widespread inflammatory responses.

5) The majority of the analyses focus on promoter regions. What about enhancers? Can the authors make a statement as to the role of promoters vs. enhancers in microglial responses to Abeta pathology?! Also, the authors compare the concordance between ATAC- and RNA-seq through correlating FC values between DEGs and DARs. A similar comparison should be made within groups, i.e. how good is the correlation of FC values between samples within the same experimental group? This will validate the robustness of the data (also see point 1).

6) For the WGCNA analysis, a module heat map showing the correlation coefficients and p-values needs to be presented to allow for interpretation.

7) Regarding the very interesting finding that VDR might drive an anti-inflammatory microglial phenotype in APOE2 cells: The IL-10RA subunit is also upregulated in APOE ko cells; is there also enrichment of VDR motifs in the DARs of these cells? Which DARs show the VDR motif? Can the authors infer which genes are expressed due to VDR activation in microglia? In the WGCNA analysis, which modules contain VDR and/or IL10-R subunits and/or VDR target genes? Which microglial subtypes express VDR/IL10-R/VDR target genes in the authors' recent single cell analysis (Mancuso et al., Nat. Neurosci, 2024)?

8) The authors conclude that their data "point toward diminished migratory capacity in APOE4 and enhanced migratory capacity in APOE2 microglia". This interpretation would be much strengthened if some evidence could be provided for such functional effects, either using analysis of tissue from xenotransplanted animals or a careful literature search.

9) The authors state in the methods section that "another APOE-KO and one APOE4 microglia sample were discarded because they did not meet QC standards". Please specify the criteria applied for this decision.

Further concerns:

10) The authors stipulate that APOE isoforms "functionally alter microglia". However, no functional assessments are performed. Please phrase more carefully, as function is only inferred from molecular profiling data, which is not always correct.

11) The authors claim that "Abeta deposition is lower in AD mouse models with APOE knocked out". While this is true, this comparison is not valid because these are animals with global APOE ko. A recent study by the Green lab showed that microglia-specific Apoe ko did not change total ABeta plaque load.

(Remarks on code availability)

Reviewer #2

(Remarks to the Author)

This manuscript investigates the effects of different APOE isoforms on the transcriptomic and epigenomic landscapes of human microglia, crucial components of the cellular response in Alzheimer's disease (AD), using a xenotransplantation model. Utilizing RNA-seq and ATAC-seq, the study reveals significant differences between APOE2, APOE3, APOE4 and APOE-KO microglia. The results indicate that APOE4 microglia exhibit heightened pro-inflammatory responses compared to APOE2, potentially contributing to increased AD risk. Additionally, APOE2 microglia exhibit upregulated cellular migration and immune responses compared to other APOE isoforms. Notably, enrichment analysis suggests that the regulatory role of the DNA binding motif of the Vitamin D Receptor (VDR) and increased anti-inflammatory signaling in APOE2 microglia contribute to its protective effects.

Previous study (Mancuso et al.) have demonstrated at the single-cell level that APOE isoforms modulate the transcriptional states of xenotransplanted microglia (xMGs). In this study, the authors examined transcriptional and epigenomic changes in xMGs of APOE2, APOE3, and APOE4 using bulk sequencing and epigenomic analysis, highlighting significant changes in genes and pathways in APOE2 microglia. However, due to the lack of functional validation, the conclusions remain descriptive and speculative, limiting our understanding on how APOE2 modulates microglia function that benefits AD pathology. Additionally, the following concerns need to be addressed to better support the conclusion:

Major Concerns:

1. Considering the transcriptomic and epigenetic sexual dimorphism in microglia, it is crucial to account for the sex of both the donor and recipient in this study. The impact on the transcriptomic, epigenetic, and functional properties of the transplanted microglia should be examined.
2. What are the factors that account for the large variation in APOE expression profiles observed within the APOE4 group and the APOE-KO group (Fig. 1b and Supplementary Fig. 2)? Is the variation due to methodological or technical issues, or are there other factors influencing APOE expression? Addressing these questions will enhance the clarity and reliability of the subsequent analysis.
3. The author suggests both pro-inflammatory and anti-inflammatory roles for APOE2, which is contradictory. To what extent do different pathways contribute to microglial functions and AD pathology, and how does APOE2 regulate these responses?
4. Although the study provides transcriptomic and epigenomic data and identifies key differentially expressed genes, such as CHCHD2 and ZNF248, the interpretation remains speculative without functional validation. How these changes translate into functional differences in microglia should be addressed by functional assays.
5. The increased chromatin accessibility in the VDR region in APOE2 microglia is intriguing. However, understanding how APOE2 enhances this accessibility compared to other isoforms will provide valuable insights. Additionally, validating the specific downstream effects of VDR signaling through knockout assays will elucidate the precise pathways involved and help determine how APOE2 regulates microglia function.

Minor Concerns:

1. Please provide FACS data and xMGs specific antibody staining images to support engraftment efficiency and specificity of xMGs.
2. Please provide the source of stem cells used to derive microglia in this study.

3. Please ensure the accuracy of graphical annotations and text (e.g., Fig. 1a CD11+ and CD4b+, lines 47 and 80 descriptions of APOE genotype, lines 346 and 347 descriptions of ZNF248 upregulation and downregulation).

(Remarks on code availability)

I did look through but do not have time to go through details or trying to install/running the application. I hope other reviewers or the editor can look into more details.

Reviewer #3

(Remarks to the Author)

Murphy et al., present a timely study profiling APOE-dependent transcriptomic and epigenetic programs in human microglia in response to A β pathogenesis. They have characterized these responses using a xenotransplantation model of human microglia in an AD mouse model with slow-progressing A β pathology. Overall, this study provides some noteworthy initial observations on the potential roles of the different APOE isoforms in microglia but requires far more controls and validation to substantiate the findings. We believe this study has missed an opportunity to harness the real power of mouse models—their pathology and age-associated behaviors, to demonstrate the importance of microglial APOE on age-associated phenotypes.

Major comments

1. The overall study design performs xenotransplantation of the different APOE genotype microglia into the APP-NL-G-F mice shortly after birth and then profiles the microglia after 12 months. It would be extremely helpful to pathologically characterize the mice after 12 months—how many microglia incorporate (does this vary by genotype), where do they reside (does this vary by genotype), are there in fact, plaques as the authors note have been shown in other studies (and does pathology change with genotype)? All these questions and more could be assessed with a pathological characterization of the endpoint.
2. The study does not xenotransplant human iPSC microglia into control mice. Do we know that the transcriptional and the ATAC seq profiles of these microglia are different from a non-AD mouse brain? This seems like a fundamental control that was not included in the study.
3. The studies only use a single iPSC line. Although we recognize that these are not easy experiments, the iPSC field has suffered from making conclusions based on a single human line. It would greatly strengthen the conclusions of the paper if these were executed (or at least validated) in multiple iPSC lines. Otherwise it is difficult to deconvolute the effects of genetic background or off target effects of the genome editing that was likely used to make the lines.
4. Due to factors that were not elaborated on by the authors, it looks like only 2 mice were used for the APOE-KO studies. This is a small N and makes it incredibly difficult to draw
5. Were the same number of cells collected for each genotype? Were there differences in microglial survival and proliferation? This could be answered by staining or reporting cell recovery per animal across the various animals.
6. The authors spend much of the paper on the bioinformatic analysis of these findings. Although these generate some intriguing hypotheses, none of these are tested or validated with an orthogonal dataset. In addition, many of these conclusions are based on changes of a single or very few genes. More evidence is needed (by comparison to published datasets or functional validation, perhaps with more than one iPSC line) to support their claims:
 - a. Line 341 (Discussion): is it right to claim “neuroprotective function” when high levels of CHCHD2 are also seen in the neutral genotype APOE3?
 - b. The authors conclude that APOE4 microglia shift to a CRM state based only on the observed upregulation of a subset of specific pro-inflammatory cytokines. More evidence should be provided by the authors to support this conclusion. For example, they could compare their transcriptomic dataset with that of human AD (APOE4/4) patients (<https://doi.org/10.1038/s41586-024-07185-7>).
 - c. The authors conclude that APOE2 microglia are protective by increasing A β clearance through enhanced proliferation and migration. However, no functional validation is provided to support this claim. Did the authors observe reduced A β pathology in APOE2 mice?
 - d. VDR regions are enriched in APOE2 compared to APOE4 microglia. Are these regions also enriched compared to APOE3 microglia? If so, this would strengthen the evidence to suggest that protection occurs via VDR-mediated pathways.
 - e. Besides IL-10, are there more anti-inflammatory cytokines that are upregulated in APOE2 compared to the rest of genotypes? It is insufficient to conclude that anti-inflammatory signaling is upregulated based on one cytokine.
 - f. Does the transcriptomic signature of APOE2 correlate with any of the reported AD-associated microglial states (i.e., DAM, HLA, CRM)? What is the evidence to claim that APOE2 microglia mount a pro-inflammatory response?
 - g. The authors observe similarities between APOE4 and KO microglia. Are these similarities captured at the pathology level in mice?
7. The study captures the role of APOE at a specific stage of AD pathogenesis, which includes high A β burden, neuroinflammation and synaptic loss, but excludes relevant human AD hallmarks like tau tangles and brain atrophy. Therefore, it is possible that the response of APOE to this additional disease hallmarks differs from the author’s observations. Similarly, APOE might exert risk or protection via different genetic responses prior to the onset of A β pathogenesis.
8. Given that microglial progenitors are injected at P2 in mice, it would be helpful that the authors express their thoughts on how they think APOE2 and APOE4 modulate disease development in this model:
 - a. Does it prevent/worsen the pathology induced by APPNL-G-F mutations?
 - b. Does it modify the brain microenvironment to prevent/promote A β pathogenesis?

9. To support their findings, it would be important to validate their observations in the mouse model (e.g., are disease phenotypes worse in APOE4 mice compared to APOE2?).

Minor comments

- Line 47: 'APOE4' is repeated.
- Line 80: 'APOE4/0' is repeated.
- Fig 1A: is 'CD4b+' in FACS schematic correct?
- Fig. 2D, 2E: it could be helpful to the reader to describe the meaning of the color scheme.
- Line 186: check grammar.
- Supplementary Tables have not been provided.
- Lines 242-243: check grammar.
- Line 261: the evidence that APOE4 microglia acquire a CRM-like state is weak, therefore the statement should be softened.
- Fig. 6D is not referred to in the manuscript.
- Lines 329-330: check grammar.
- Methods: specify what age were the mice when microglia were harvested.
- Discussion: is it accurate to refer to this xenotransplantation model as "late-onset AD" (see DOI: 10.1002/alz.13840), because the pathology is induced via mutations in APP (which cause familial AD)?

(Remarks on code availability)

Reviewer #4

(Remarks to the Author)

In this manuscript, Murphy and colleagues use transcriptomics and open chromatin analysis to describe potential differences in gene expression and putative gene regulation between different Apoe isoforms. This work features the use of iPSC cells which have been engrafted into a humanized Alzheimer's disease model; this methodology would allow in vivo functional interrogation of human genetic mutations with enhanced translational applicability outside of traditional mouse models. The authors found that the APOE4 isoform was de-enriched in genes associated with the DAM subset, which has been shown to be protective in mouse AD models. Additionally, APOE4 microglia upregulate pro-inflammatory cytokines, reflecting past mouse studies with humanized APOE4 alleles. However, the authors also found that some selective pro-inflammatory cytokines were upregulated in the APOE2 isoform, which is known to be protective against AD, raising future questions of how the inflammatory status of microglia may have temporal beneficial and antagonistic effects on AD progression. Lastly, the authors also found enrichment for a motif matching the Vitamin D receptor, suggesting the possibility of differential effects of vitamin D signaling dependent on APOE isoforms. This study appears to be an additional follow-up the original humanized mouse engraftment studies completed by Fattorelli et al., Nat. Protocols 2021. The authors made several observations that are in-line with previous publications also investigating APOE isoforms as well as new observations into potential gene regulation mechanisms. Despite using several APOE isoforms, there is a lack of strong inferential conclusions made about the pattern of gene expression or putative mechanisms being changed across the APOE isoforms.

Major Comments:

1. The manuscript would be stronger and more applicable to translational studies if the iPSC microglia were not null in the other allele. These iPSC lines are available in Fattorelli et al. Can the authors explain why only 1 allele was used and the other was null, especially given that APOE4/4 significantly raises risk over APOE4/0.
2. Are there any inferences that can be made about APOE2 in relation to the previously published single-cell data?
3. Are there any differential transcription factor motifs enriched for open chromatin regions that are differential between APOE2 and APOE4 and all of the other comparisons?
4. The manuscript would strongly benefit from reorganization as the premise is a comparison between the isoforms. However, in some sections, some isoforms are mentioned while others are neglected and how the results related to an APOE null, etc. It is highly difficult to synthesize what are the major findings of this manuscript is that is unique from what has already been published.

Minor Comments:

1. Line 70 – please specify what the "human-specific microglial response" is? The way it is written is ambiguous.
2. Line 80 – typo: Apoe3 is missing
3. Figure 1: CD11+, CD4b+ (wrong label)
4. Figure 1: figure legends can use more detail that match the text (e.g. number of samples with what assay)
5. SFig2 is the same as Fig1b.

(Remarks on code availability)

Code is appropriate; data files are provided and code is appropriately annotated on a figure basis.

Version 1:

Reviewer comments:

Reviewer #1

(Remarks to the Author)

The authors have addressed all of my original concerns and have significantly improved the manuscript overall. However, the FACS gating strategy that is now included in Suppl. Figure 9 raises new concerns, as it differs from the strategy reported in Mancuso et al., 2024, where CD11b+ cells were gated first and a distinction was then made between mouse CD45+ and human CD45+ cells. Why was a different strategy chosen here? Most concerning, the sorted human CD45+ cells are not clearly positive for CD11b, indicating that the cells analyzed may not (all) be microglia. The authors need to demonstrate that the sorted cells are highly enriched for microglia vs. other immune cell markers to ascertain the identity of the population analyzed.

(Remarks on code availability)

I have not assessed the code.

Reviewer #2

(Remarks to the Author)

The authors have adequately addressed my concerns. One minor concern remains which relates to Fig. 4. Because of the lower expression level of APOE4 shown in Figure 1b, it is critical to use adequate and consistent statistical methods. Specifically, although the expression levels of chemokines and cytokines in the APOE4 group are generally higher than those in the E2 and E3 groups, the within-group data exhibit high heterogeneity. As such, when comparing the expression levels of specific genes among the APOE2, APOE3, APOE4, and KO groups, as shown in Figures 4c, d, and e, applying FDR correction appears unnecessary. If a one-way ANOVA or the same statistical test as in Figure 1b is applied, are the differences between the groups still statistically significant? As these are related to a major conclusion of this work, it is critical to apply more rigor to these analyses.

(Remarks on code availability)

Reviewer #3

(Remarks to the Author)

We appreciate the authors' response. The authors have gone through all comments and addressed each point.

The two primary pieces of experimental data the authors added were 1) phagocytosis assays by microglia and 2) FACS analysis of cell abundance. Many of the other validation cases were through other's work or referring to past work in Mancuso et al.

Although interesting, the myelin phagocytosis assay is used as a "parallel" to chemotaxis--this seems a bit of a stretch. Phagocytosis and chemotaxis are different things and rely on different pathways. Therefore, using phagocytosis assays to functionally validate chemotaxis does not seem sound.

I realize the difficulty of the xenotransplantation experiments. However, the authors seem to continually refer to Mancuso et al and other pieces of work to support their sequencing datasets. This renders the current paper more of a data resource rather than a study with strong validation and a better fit for another journal.

(Remarks on code availability)

Unfortunately, I do not have the expertise to review the code.

Reviewer #4

(Remarks to the Author)

The authors have adequately addressed the comments. The additional phagocytosis data adds to the Vitamin D receptor finding. Clarifications were also made regarding methodology and exclusions.

(Remarks on code availability)

Version 2:

Reviewer comments:

Reviewer #1

(Remarks to the Author)

The authors have addressed my remaining concern. Congratulations on a very interesting study!

(Remarks on code availability)

Reviewer #2

(Remarks to the Author)

The authors have adequately addressed my concerns and I have no further comments.

(Remarks on code availability)

I do not have expertise to review the code.

Reviewer #3

(Remarks to the Author)

The authors clarified their writing and figure relating to phagocytosis. This "omics" work generates intriguing hypotheses that will need future validation. I have no new comments.

(Remarks on code availability)

n/a

Responses to the reviewers for NCOMMS-24-44479-T “The APOE isoforms differentially shape the transcriptomic and epigenomic landscapes of human microglia in a xenotransplantation model of Alzheimer’s disease”

We would like to thank the reviewers for offering their time and expertise to review our manuscript. Overall, the four reviewers raised many important and thought-provoking questions. We have amended the manuscript to include new experiments, analyses and discussions and believe these have strengthened the overall impact and quality of the paper. Please see our point-by-point response to the reviewers below.

Reviewer #1 (Remarks to the Author):

This study by Murphy et al. utilizes the xenotransplantation model developed by the de Strooper lab to study epigenetic regulation of human microglial phenotypes driven by different APOE isoforms in the context of Abeta pathology, reporting some interesting findings with regards to cellular response regulation in APOE2 and APOE4 vs. APOE3 cells. Through comparison with APOE ko cells, their study also allows for interpretation with regard to whether cellular phenotypes are driven by loss of function. While the paper certainly adds novel insights into how APOE isoforms cell-autonomously affect microglial responses, some concerns need to be addressed before publication.

Major concerns:

1) The authors report exclusion of two APOE ko samples that showed no reduction in APOE gene expression. While I applaud the authors for their honesty, this raises concerns regarding the identity of the transplanted iPS-derived microglia. This is particularly the case, because the APOE4 group also shows two samples with conspicuously higher APOE expression levels as well as two samples that cluster closer to APOE ko and APOE2 cells in the PCA and hierarchical clustering analyses. The genotype of all samples needs to be verified (using the existing sequencing data or qPCR) to ensure that these cells are indeed the correct genotype. Otherwise, the data cannot be interpreted.

Along the same line of thought, Figure 1c shows some ATAC reads within the gene body of the APOE gene in APOE ko samples and very clearly in the promoter region. Is the promoter region still present in this knockout line? Please provide more details.

We thank the reviewer for this question and the chance for clarification. The genotypes of the samples were confirmed by western blot prior to the study. However, as we detected some APOE expression in the KO we chose to err on the side of caution and remove these samples from downstream analyses. We have now also re-genotyped the samples using Sanger sequencing and PCR:

rs429358
ATGGAGGACGTGCGCGGCCGCCTGGTG

ATGGAGGACGTGCGCGGCCGCCTGGTG
ATGGAGGACGTGCGCGGCCGCCTGGTG

ATGGAGGACGTGCGCGGCCGCCTGGTG
ATGGAKGACGTGTGCGGACGCCTGGTG

ATGGAKGACGTGTGCGGACGCCTGGTG
ACTYAAACACGGTCTMTGTAAKGYGAA

ACTYAAACACGGTCTMTGTAAKGYGAA

BIONi010-C4
(APOE ϵ_4)

BIONi010-C2
(APOE ϵ_3)

BIONi010-C6
(APOE ϵ_2)

BIONi010-C3
(APOE-KO)

rs7412
GACCTGCAGAAGCGCCTGGCAGTGTAC

GACCTGCAGAAGCGCCTGGCAGTGTAC
GACCTGCASAAGCGCCTGGCAKTGTAC

GACCTGCASAAGCGCCTGGCAKTGTAC
GACCTGCASAARTGCTGACATTGTAC

GACCTGCASAARTGCTGACATTGTAC
GTGMCATTTGTTGAATGTASGACGTAK

GTGMCATTTGTTGAATGTASGACGTAK

M – ladder

1 – APOE2

2 – APOE3

3 – APOE4

4 – APOE-KO

5 – Blank PCR

6 - ladder

We have also re-run analyses where we make inferences about the knockout using all five KO samples, and confirmed that the results remain unchanged (please see example figures below where all five KO samples have been included). In addition, **Supplementary Fig. 2** (also shown below) shows that even with all KO samples included, APOE expression remains significantly lower in the KO samples.

Regarding the reviewer's comment on the chromatin accessibility signature in the APOE gene body for the knockout, we would like to point out that the APOE-KO is introduced by a frameshift mutation. The non-functional transcript can still be transcribed and we would expect to see open chromatin signatures in the promoter (and gene body) region.

2) No supplementary tables were submitted, preventing assessment of some of the results.

Apologies, all supplementary tables are available on zenodo at doi: 10.5281/zenodo.12516685 as outlined in the data availability statement. For the revised submission we have also submitted the supplementary tables as individual files to the journal.

3) All experiments are performed in APP transgenic animals, so it is not possible to conclude which molecular alterations are due to the different APOE isoforms per se or due to the Abeta pathology x APOE isoform interaction. This needs to be made clear throughout the text.

We agree with the reviewer, and while we haven't looked into this in our study, Mancuso and colleagues (2024) showed that certain microglial states were significantly enriched in response to amyloid beta pathology when compared to wild type (Mancuso et al. 2024). Thus suggesting that at least some responses are due to Abeta pathology x APOE isoform interaction.

4) The authors report only few DARs compared to the number of DEGs. How was the fold-change cut-off chosen for DAR detection? Under conditions of chronic inflammation and because the samples were analyzed in bulk with only a subset of cells responding to Abeta pathology, one might expect a lower FC than found in equivalent experiments during acute and widespread inflammatory responses.

The reviewer makes a good point about only a subset of cells responding to the amyloid beta pathology. However, this issue of subsets of cells responding should affect the gene expression data in the same way. We call all peaks and transcripts with $FDR < 0.05$ as significant, independent of $\log FC$, because we believe that selecting a $\log FC$ can be arbitrary. The lower number of DARs could be partly due to a power issue whereby it is more difficult to detect DARs than DEGs due to the higher number of chromatin accessibility peaks (167,951) versus genes (17,756) that undergo multiple testing correction. Taking APOE2 vs APOE3 as an example, we have 474 significant genes out of 17,756. In the DA analysis, we have 40 significant peaks out of 167,951. To test this, we randomly subset the ATAC-seq peaks to the same number as genes. In this scenario the number of DARs increased to 91, indicating that multiple testing-related power issues are at least partially responsible for differences in the number of DARs vs DEGs.

5) The majority of the analyses focus on promoter regions. What about enhancers? Can the authors make a statement as to the role of promoters vs. enhancers in microglial responses to Abeta pathology?! Also, the authors compare the concordance between ATAC- and RNA-seq through correlating FC values between DEGs and DARs. A similar comparison should be made within groups, i.e. how good is the correlation of FC values between samples within the same experimental group? This will validate the robustness of the data (also see point 1).

We completely agree with the reviewer that enhancers should also be considered, and are arguably particularly interesting in the microglial response to Abeta. In **Supplementary Fig. 1e** we show the distribution of ATAC-seq peak to genomic feature annotation, which includes promoters and distal elements such as enhancers. All our differential analyses were conducted on all ATAC-peaks, comprising the promoter and enhancer regions. However, for the direct comparisons across the two assays (RNA-seq and ATAC-seq), we chose to focus on promoters, to be more confident regarding the annotation of peaks to genes. For promoter peaks this is straightforward, while an enhancer peak may not necessarily regulate the closest gene. The reviewers' comments prompted us to additionally correlate the $\log FC$ between all ATAC-seq peaks and the $\log FC$ of expression of the genes they are annotated to (if $FDR < 0.05$ for the gene in the RNA-seq data), and these still show a strong correlation. Please see below for an example when comparing APOE4 to APOE3. In terms of correlating the samples, this is shown in **Supplementary Fig. 1c,d** for both the RNA-seq and ATAC-seq.

6) For the WGCNA analysis, a module heatmap showing the correlation coefficients and p-values needs to be presented to allow for interpretation.

We thank the reviewer for their suggestion and have added a WGCNA module heatmap in **Fig. 5a**. The turquoise module and the dark green module correspond to the proliferation and migration module and the immune response module discussed in the manuscript (up in APOE2 after differential module expression analysis), respectively. Please see below for the module heatmap.

7) Regarding the very interesting finding that VDR might drive an anti-inflammatory microglial phenotype in APOE2 cells: The IL-10RA subunit is also upregulated in APOE ko cells; is there also enrichment of VDR motifs in the DARs of these cells? Which DARs show the VDR motif? Can the authors infer which genes are expressed due to VDR activation in microglia? In the WGCNA analysis, which modules contain VDR and/or IL10-R subunits and/or VDR target genes? Which microglial subtypes express VDR/IL10-R/VDR target genes in the authors' recent single cell analysis (Mancuso et al., Nat. Neurosci, 2024)?

We thank the reviewer for these interesting suggestions. The VDR motif is not enriched in the APOE-KO, suggesting that this may be a protective gain of function in the APOE2 isoform. 10 out of the top 100 upregulated peaks in APOE2-expressing microglia were enriched for the VDR motif and we can share a .csv file of these in the supplementary. The ME1 (dark green) module is enriched for upregulated VDR target genes, and this module was also identified as upregulated in APOE2-expressing microglia - see figure below, which has been added as **Supplementary Fig. 8** to the manuscript. The ME1 module was characterised by biological pathways associated with immune responses. The ME2 module is also enriched for upregulated VDR target genes, however, this module was not differentially expressed across the APOE groups (FDR < 0.05). We have added these new results to the manuscript. VDR target genes were not enriched in any of the microglia clusters identified in Mancuso et al. (2024). IL10RA is upregulated in tCRM, RM, and DAM.

8) The authors conclude that their data “point toward diminished migratory capacity in APOE4 and enhanced migratory capacity in APOE2 microglia”. This interpretation would be much strengthened if some evidence could be provided for such functional effects, either using analysis of tissue from xenotransplanted animals or a careful literature search.

As helpfully suggested by the reviewer, a literature search supports our interpretation of APOE4-expressing microglia having defects in their migratory capacity:

- 1) Migration is decreased in APOE4 microglia-like cells (Konttinen et al. 2019).
- 2) Genes associated with microglial migration are upregulated in APOE2 microglia (Wang et al. 2022).
- 3) APOE4 microglia have reduced motility and responsiveness to ATP, a chemotactic cue of neuronal damage, and amyloid beta (Sepulveda et al. 2024).

These references have now been included in the discussion.

In addition, we have added new functional experimental work to the manuscript: If APOE2-expressing microglia have enhanced migratory capacity, we hypothesised this would be paralleled by increased phagocytic activity and have now shown this using a phagocytosis assay with *E. coli* and myelin. The figures below have been included in the revised manuscript (**Fig. 6**).

iPS cell microglia harbouring different *APOE* allelic variants display differences in the uptake of *E. Coli* pHrodo and fluorescent myelin particles. **a,b** Representative images and quantification of pHrodo *E. Coli* particles (100 μ g/ml) uptake by *APOE2*, *APOE3*, *APOE4* and *APOE-KO* iPS cell-derived microglia. Each data point represents an independent well. An average of 1236 +/- 87 cells were quantified per well. **c,d** Representative images and quantification of PKH67 labelled myelin (200 μ g/ml) uptake by *APOE2*, *APOE3*, *APOE4* and *APOE-KO* iPS cell-derived microglia. Each data point represents an independent well. An average of 211 +/- 9 cells were quantified per well. Statistical analysis was performed using ANOVA, with Bonferroni correction for multiple comparisons; * $p < 0.05$, ** $p < 0.01$ and *** $p < 0.001$. **e** Barplot showing the overrepresentation of genes upregulated in *APOE2* microglia in a set of genes upregulated in phagocytic microglia responsive to A β plaques (Grubman et al. 2021). The dashed line represents the significance threshold after FDR correction ($p < 0.05$).

9) The authors state in the methods section that “another APOE-KO and one APOE4 microglia sample were discarded because they did not meet QC standards”. Please specify the criteria applied for this decision.

Apologies for not making this clear, both samples were removed due to having less than 9 million reads after QC. This has now been added to the revised manuscript.

Further concerns:

10) The authors stipulate that APOE isoforms “functionally alter microglia”. However, no functional assessments are performed. Please phrase more carefully, as function is only inferred from molecular profiling data, which is not always correct.

The reviewer makes a good point here, we have changed “functionally alter microglia” to “change regulatory patterns that can be linked back to biological functions”.

11) The authors claim that “Abeta deposition is lower in AD mouse models with APOE knocked out”. While this is true, this comparison is not valid because these are animals with global APOE ko. A recent study by the Green lab showed that microglia-specific Apoe ko did not change total ABeta plaque load.

We thank the reviewer for carefully pointing this out, we have removed the sentence “A β deposition is lower in AD mouse models with APOE knocked out” in the revised manuscript.

Reviewer #2 (Remarks to the Author):

This manuscript investigates the effects of different APOE isoforms on the transcriptomic and epigenomic landscapes of human microglia, crucial components of the cellular response in Alzheimer's disease (AD), using a xenotransplantation model. Utilizing RNA-seq and ATAC-seq, the study reveals significant differences between APOE2, APOE3, APOE4 and APOE-KO microglia. The results indicate that APOE4 microglia exhibit heightened pro-inflammatory responses compared to APOE2, potentially contributing to increased AD risk. Additionally, APOE2 microglia exhibit upregulated cellular migration and immune responses compared to other APOE isoforms. Notably, enrichment analysis suggests that the regulatory role of the DNA binding motif of the Vitamin D Receptor (VDR) and increased anti-inflammatory signaling in APOE2 microglia contribute to its protective effects.

Previous study (Mancuso et al.) have demonstrated at the single-cell level that APOE isoforms modulate the transcriptional states of xenotransplanted microglia (xMGs). In this study, the authors examined transcriptional and epigenomic changes in xMGs of APOE2, APOE3, and APOE4 using bulk sequencing and epigenomic analysis, highlighting significant changes in genes and pathways in APOE2 microglia. However, due to the lack of functional validation, the

conclusions remain descriptive and speculative, limiting our understanding on how APOE2 modulates microglia function that benefits AD pathology. Additionally, the following concerns need to be addressed to better support the conclusion:

Major Concerns:

1. Considering the transcriptomic and epigenetic sexual dimorphism in microglia, it is crucial to account for the sex of both the donor and recipient in this study. The impact on the transcriptomic, epigenetic, and functional properties of the transplanted microglia should be examined.

We agree with the reviewer that it is extremely important to account for sexual dimorphism in these kinds of studies, however, we point out that the donor and host sex are fully controlled as the donor iPSC line is male and all host mice are male. We have added statements in the methods section of the revised manuscript to make this more clear. And while investigation of sex specific effects of the host and donor would be of great interest, it goes beyond the scope of this study.

2. What are the factors that account for the large variation in APOE expression profiles observed within the APOE4 group and the APOE-KO group (Fig. 1b and Supplementary Fig. 2)? Is the variation due to methodological or technical issues, or are there other factors influencing APOE expression? Addressing these questions will enhance the clarity and reliability of the subsequent analysis.

The reviewer has raised an interesting question. Two studies, referenced in the manuscript, have shown that APOE expression in APOE4 astrocytes and microglia is lower than other isoforms and this may contribute to the variability in this isoform. In terms of the KO, we wanted to be very cautious with samples showing higher APOE expression and therefore removed these from our analyses.

3. The author suggests both pro-inflammatory and anti-inflammatory roles for APOE2, which is contradictory. To what extent do different pathways contribute to microglial functions and AD pathology, and how does APOE2 regulate these responses?

We understand the reviewers' concern here, however, we would argue that it is still unclear to what extent pro-inflammatory and anti-inflammatory responses are protective or pathogenic in AD. In addition, whether a cytokine is pro- or anti-inflammatory is context-dependent. While we believe APOE2 is largely driving an anti-inflammatory response as part of its protective functions, we found that it also upregulates the expression of TNF family genes, TNF itself being pro-inflammatory. Such discrepancies can be attributed to the highly pleiotropic nature of APOE. Out of 40 TNF family genes, eight were upregulated in APOE2 vs APOE4 (TNFSF12, TNFRSF14, TNFRSF21, TNFRSF1B, TNFSF8, TNFRSF13C, TNFSF12-TNFSF13, TNFRSF1A). When comparing APOE2 vs APOE3, only two genes were upregulated (TNFRSF25, TNFRSF21). The top-upregulated TNF family gene in APOE2 vs APOE4 was TNFSF12. In an ALS mouse model, TNFSF12 induces neuronal death but also promotes microgliosis (Bowerman et al. 2015). One could argue that to some extent these functions are beneficial, but chronically can be pathogenic. We have now elaborated on these points in the manuscript.

4. Although the study provides transcriptomic and epigenomic data and identifies key differentially expressed genes, such as CHCHD2 and ZNF248, the interpretation remains speculative without functional validation. How these changes translate into functional differences in microglia should be addressed by functional assays.

CHCHD2 has previously been characterised as a cellular migration-promoting gene (Seo et al. 2010; Wei et al. 2015), and this aligns with the enrichment of pathways associated with chemotaxis/migration. We speculated that if APOE2-expressing microglia have enhanced migratory capacity, this would be paralleled by increased phagocytic activity. We performed a new series of functional experiments, using a phagocytosis assay with E. coli and myelin. These confirmed our hypothesis and have now been included in our updated manuscript. Please see the response to comment #8 from Reviewer 1, and the new **Fig. 6** in the revised manuscript for more details.

5. The increased chromatin accessibility in the VDR region in APOE2 microglia is intriguing. However, understanding how APOE2 enhances this accessibility compared to other isoforms will provide valuable insights. Additionally, validating the specific downstream effects of VDR signaling through knockout assays will elucidate the precise pathways involved and help determine how APOE2 regulates microglia function.

For validation of functional effects please see our response to the previous question as well as the response to comment #8 by reviewer 1, which describe new functional experimental work using a phagocytosis assay with E. coli and myelin. As far as the direct functional link between APOE2 and VDR binding goes, we agree that this is an interesting question, but unfortunately exceeds the scope of this manuscript. We hope to expand further on this question in future research projects.

Minor Concerns:

1. Please provide FACS data and xMGs specific antibody staining images to support engraftment efficiency and specificity of xMGs.

We thank the reviewer for pointing this out. Please see below a representative image of the gating strategy for the sorting of human microglia. Cells were pre-gated using a mouse/human cross-reactive CD11b antibody, and further separated using mouse and human-specific anti-CD45 antibodies. The sorted population is outlined and depicted as hMG. This has been now included as a supplementary figure in the manuscript (**Supplementary Fig. 7**).

2. Please provide the source of stem cells used to derive microglia in this study.

Please see below a table containing the source of stem cells.

Name of line	Genotype	Source	Citation
BIONi010-C-2	APOE e3/KO	Bioneer, EBiSC	RRID:CVCL_I181
BIONi010-C-3	APOE KO/KO	Bioneer, EBiSC	RRID:CVCL_I182
BIONi010-C-4	APOE e4/KO	Bioneer, EBiSC	RRID:CVCL_I183
BIONi010-C-6	APOE e2/KO	Bioneer, EBiSC	RRID:CVCL_I185

3. Please ensure the accuracy of graphical annotations and text (e.g., Fig. 1a CD11+ and CD4b+, lines 47 and 80 descriptions of APOE genotype, lines 346 and 347 descriptions of ZNF248 upregulation and downregulation).

Thank you to the reviewer for spotting this. It should be hCD45+.

Reviewer #2 (Remarks on code availability):

I did look through but do not have time to go through details or trying to install/running the application. I hope other reviewers or the editor can look into more details.

Reviewer #3 (Remarks to the Author):

Murphy et al., present a timely study profiling APOE-dependent transcriptomic and epigenetic programs in human microglia in response to A β pathogenesis. They have characterized these responses using a xenotransplantation model of human microglia in an AD mouse model with slow-progressing A β pathology. Overall, this study provides some noteworthy initial observations on the potential roles of the different APOE isoforms in microglia but requires far more controls and validation to substantiate the findings. We believe this study has missed an opportunity to harness the real power of mouse models—their pathology and age-associated behaviors, to demonstrate the importance of microglial APOE on age-associated phenotypes.

Major comments

1. The overall study design performs xenotransplantation of the different APOE genotype microglia into the APP-NL-G-F mice shortly after birth and then profiles the microglia after 12 months. It would be extremely helpful to pathologically characterize the mice after 12 months—how many microglia incorporate (does this vary by genotype), where do they reside (does this vary by genotype), are there in fact, plaques as the authors note have been shown in other studies (and does pathology change with genotype)? All these questions and more could be assessed with a pathological characterization of the endpoint.

Please see the response regarding the number of engrafted microglia per animal and genotype in the response to this reviewer's comment #5, where we display the numbers of cells recovered as well as graft efficiency per mouse. We observed graft efficiencies between 60-90% across all the APOE genotypes (see figure below). We have previously characterized this model thoroughly across multiple lines harbouring different genotypes and observed a vast colonization of human microglia in the cortex, hippocampus, striatum and basal ganglia and olfactory bulb (Fattorelli and Martinez-Muriana et al. Nature Protocols 2021).

We have previously characterized the response of APOE2/3/4/KO human microglia transplanted in the mouse brain, as well as their association with amyloid beta plaques in the AppNL-G-F mouse model at 6-7 months of age. We previously confirmed all APOE genotypes display reactive states to amyloid beta plaques, with a partial impairment in the APOE4 and APOE-KO genotypes (Mancuso, Fattorelli and Martinez-Muriana et al. Nat Neurosci 2024), consistent with the data we present here. Plaques were observed across all genotypes, however, they were not quantified and so we cannot comment on differential pathology.

2. The study does not xenotransplant human iPSC microglia into control mice. Do we know that the transcriptional and the ATAC seq profiles of these microglia are different from a non-AD mouse brain? This seems like a fundamental control that was not included in the study.

As described in response to Reviewer 1, this was done in Mancuso et al (2024) where they showed that some microglial states were significantly enriched specifically in response to amyloid beta pathology when compared to wild type (Mancuso et al. 2024).

3. The studies only use a single iPSC line. Although we recognize that these are not easy experiments, the iPSC field has suffered from making conclusions based on a single human line. It would greatly strengthen the conclusions of the paper if these were executed (or at least validated) in multiple iPSC lines. Otherwise it is difficult to deconvolute the effects of genetic background or off target effects of the genome editing that was likely used to make the lines.

We completely agree with the reviewer about this issue in iPSC studies. In Mancuso et al (2024), where the same microglia xenotransplantation model was used, iPSC lines from two different genetic backgrounds were included. The authors did not observe any differences between these two genetic backgrounds at the single-cell gene expression level. In light of this, we opted to use a single isogenic iPSC line for this study. Importantly, this still means that our between-genotype analyses weren't confounded by differences in genetic background. Unfortunately, repeating all experiments from this study in an additional line would take at least 18 months and would be beyond the scope of the study.

4. Due to factors that were not elaborated on by the authors, it looks like only 2 mice were used for the APOE-KO studies. This is a small N and makes it incredibly difficult to draw

Apologies for not making this clearer. Two APOE-KO samples were excluded due to having higher APOE expression in the RNA-seq data, so we also excluded these samples from the ATAC-seq analysis. The third KO sample excluded from the ATAC-seq analysis was discarded due to having a low number of reads (less than 9 million). We have now made this clearer in the revised manuscript. Importantly, we only drew conclusions from the KO for the RNA-seq data as here we had three samples. Nothing is concluded for the ATAC-seq from the KO. However, we have also re-run analyses where we make inferences about the knockout using all five KO samples, and confirmed that these remain unchanged (see response to comment #1 by reviewer 1). In addition, **Supplementary Fig. 2** shows that even with all KO samples included, APOE expression remains significantly lower in the KO samples.

5. Were the same number of cells collected for each genotype? Were there differences in microglial survival and proliferation? This could be answered by staining or reporting cell recovery per animal across the various animals.

Please see the table below with the numbers of cells collected from each mouse, and used for the different downstream molecular biology techniques (including CUT&Tag experiments not included in this manuscript). Based on these numbers as well as the percentage in engraftment efficiency, we have no reason to believe that there are major changes in survival or proliferation, and therefore we opted for not investigating this further.

ID	Genotype cells	ATAC-seq	RNA-seq	CUT&Tag	Total
MG539	APOE2	100000	200000	200000	500000
MG540	APOE2	100000	200000	200000	500000
MG543	APOE2	100000	200000	200000	500000
MG544	APOE2	100000	200000	200000	500000
MG545	APOE2	100000	200000	200000	500000
MG435	APOE3	100000	200000	200000	500000
MG439	APOE3	100000	200000	200000	500000
MG345	APOE3	100000	200000	200000	500000
MG346	APOE3	100000	200000	200000	500000
MG347	APOE3	100000	200000	200000	500000
MG553	APOE4	100000	200000	200000	500000
MG554	APOE4	100000	200000	200000	500000
MG655	APOE4	100000	200000	200000	500000
MG656	APOE4	100000	160000	160000	420000
MG657	APOE4	100000	160000	160000	420000
MG754	APOE-KO	100000	200000	200000	500000
MG755	APOE-KO	100000	200000	200000	500000
MG243	APOE-KO	100000	160000	160000	420000
MG244	APOE-KO	100000	110000	110000	320000
MG614	APOE-KO	100000	185000	185000	470000

6. The authors spend much of the paper on the bioinformatic analysis of these findings. Although these generate some intriguing hypotheses, none of these are tested or validated with an orthogonal dataset. In addition, many of these conclusions are based on changes of a single or very few genes. More evidence is needed (by comparison to published datasets or functional validation, perhaps with more than one iPSC line) to support their claims:

a. Line 341 (Discussion): is it right to claim “neuroprotective function” when high levels of CHCHD2 are also seen in the neutral genotype APOE3?

We agree with the reviewer and have changed this to say that it may be a loss of normal function rather than being neuroprotective.

b. The authors conclude that APOE4 microglia shift to a CRM state based only on the observed upregulation of a subset of specific pro-inflammatory cytokines. More evidence should be provided by the authors to support this conclusion. For example, they could compare their transcriptomic dataset with that of human AD (APOE4/4) patients (<https://doi.org/10.1038/s41586-024-07185-7>).

We thank the reviewer for flagging this and realise we weren't clear in what we were trying to say. We say that “Furthermore, in response to A β , APOE4 microglia shift to the CRM state rather than HLA¹⁶, referencing here Mancuso et al (2024) where they used the same microglia xenotransplantation model but with single-cell sequencing. We have now reworded this to make it more clear: “Mancuso and colleagues (Mancuso et al. 2024) showed that in response to A β , APOE4 microglia shift to the CRM state rather than HLA.”

c. The authors conclude that APOE2 microglia are protective by increasing A β clearance through enhanced proliferation and migration. However, no functional validation is provided to support this claim. Did the authors observe reduced A β pathology in APOE2 mice?

We have now added functional validation data showing that APOE2-expressing microglia have enhanced phagocytic activity (please see the figure added below the comment #8 made by reviewer 1 / **Fig. 6** in the revised manuscript). Although we didn't use amyloid beta in the phagocytosis assay, independent studies have shown that APOE2 has higher efficiency in clearing amyloid beta pathology when compared to APOE3 in the mouse brain (Deane et al. 2008; Castellano et al. 2011). APOE2-expressing macrophages derived from transgenic mice with an APP mutation were also more efficient at degrading amyloid beta than both APOE3 and APOE4-expressing macrophages (Zhao et al. 2009). This is now included in the discussion.

d. VDR regions are enriched in APOE2 compared to APOE4 microglia. Are these regions also enriched compared to APOE3 microglia? If so, this would strengthen the evidence to suggest that protection occurs via VDR-mediated pathways.

VDR regions are enriched in APOE2 vs APOE3 microglia, and not when compared to APOE4. To strengthen our finding we also looked at VDR target gene expression. If there is increased binding of VDR in APOE2 microglia then you would expect a higher expression of VDR target

genes in APOE2-expressing microglia, and this was the case when compared to both APOE4 and APOE3 (**Fig. 7b**).

e. Besides IL-10, are there more anti-inflammatory cytokines that are upregulated in APOE2 compared to the rest of genotypes? It is insufficient to conclude that anti-inflammatory signaling is upregulated based on one cytokine.

We discuss IL-10 but also CXCL16 in **Fig. 4f** which is upregulated in APOE2 microglia and has been characterised as a microglial anti-inflammatory cytokine in the context of brain tumours. However, as we only have two specific examples we have now removed the line: “suggesting increased anti-inflammatory signalling in the *APOE2* isoform.”

f. Does the transcriptomic signature of APOE2 correlate with any of the reported AD-associated microglial states (i.e., DAM, HLA, CRM)? What is the evidence to claim that APOE2 microglia mount a pro-inflammatory response?

We thank the reviewer for this question. Yes, as seen in **Fig. 4a**, genes upregulated in APOE2 microglia (see downregulated in E4vsE2 and upregulated in E2vsE3) are enriched within DAM and HLA, states which are posited to be protective. There is no enrichment of APOE2 DEGs in CRM, suggesting that they do not shift to a pro-inflammatory state. The only evidence to propose that microglia mount a pro-inflammatory response is the upregulation of certain TNF family genes. However, as described in response to reviewer 2, whether these TNF family genes are pro-inflammatory is context dependent. If they are pro-inflammatory, it also remains unclear to what extent pro-inflammatory responses are protective versus pathogenic in AD. We have revised the language throughout the manuscript to ensure we describe the specific activation of TNF family genes rather than a general pro-inflammatory response.

g. The authors observe similarities between APOE4 and KO microglia. Are these similarities captured at the pathology level in mice?

While the reviewer is correct, we see a stronger overlap between APOE2 and KO microglia. In terms of pathology, a study in which APOE was knocked out specifically in microglia, there was no effect on amyloid beta levels (Henningfield et al. 2022).

7. The study captures the role of APOE at a specific stage of AD pathogenesis, which includes high A β burden, neuroinflammation and synaptic loss, but excludes relevant human AD hallmarks like tau tangles and brain atrophy. Therefore, it is possible that the response of APOE to this additional disease hallmarks differs from the author’s observations. Similarly, APOE might exert risk or protection via different genetic responses prior to the onset of A β pathogenesis.

As suggested by the reviewer, it is reasonable to assume that other hallmarks would likely elicit different responses. For instance, work from the lab of Beth Stevens showed that exposing iPSC-derived microglia to different brain substrates had an impact on their transcriptome (Dolan et al. 2023). Therefore, throughout the manuscript, we aimed to make it clear we were investigating the

response to amyloid beta pathology, and not any other disease-specific hallmarks. In response to the last point “APOE might exert risk or protection via different genetic responses prior to the onset of A β pathogenesis”, this is also a fair assumption and we have now included this in the limitations section of the discussion.

8. Given that microglial progenitors are injected at P2 in mice, it would be helpful that the authors express their thoughts on how they think APOE2 and APOE4 modulate disease development in this model:

a. Does it prevent/worsen the pathology induced by APPNL-G-F mutations?

b. Does it modify the brain microenvironment to prevent/promote A β pathogenesis?

9. To support their findings, it would be important to validate their observations in the mouse model (e.g., are disease phenotypes worse in APOE4 mice compared to APOE2?).

These are very good and challenging questions raised by the reviewer. Based on our new functional validation data that APOE2-expressing microglia have enhanced phagocytic activity, we speculate that amyloid beta pathology is likely reduced in this isoform. This is supported by previous studies that report a lower accumulation of myelin debris in APOE2-TR mice (Wang et al. 2022) and increased clearance of amyloid beta from the interstitial fluid of mice expressing the APOE2 isoform (Castellano et al. 2011). Referring to point 8b, there is not much known about how and whether this would happen and unfortunately. We are keen to investigate this in more detail in future projects. Setting up relevant functional experiments in the xenotransplantation model would take an additional 18 months and is unfortunately beyond the scope of this study.

Minor comments

- Line 47: ‘APOE4’ is repeated.

This has been corrected in the manuscript.

- Line 80: ‘APOE4/0’ is repeated.

This has been corrected in the manuscript.

- Fig 1A: is ‘CD4b+’ in FACS schematic correct?

Thank you to the reviewer for spotting this. It is indeed incorrect and should be hCD45+.

- Fig. 2D, 2E: it could be helpful to the reader to describe the meaning of the color scheme.

The colour scheme represents the strength of the correlation according to the p-value, with red indicating a stronger correlation and blue weaker. We have added a description to the figure legend.

- Line 186: check grammar.

Thanks to the reviewer for spotting this, it is now changed to: “Notably, our analysis revealed consistent epigenomic and transcriptomic responses across microglia expressing the different *APOE* isoforms (**Supplementary Fig. 5**).”

- Supplementary Tables have not been provided.

Apologies, all supplementary tables are available on zenodo at doi: 10.5281/zenodo.12516685 as outlined in the data availability statement. For the revised submission we have also submitted the supplementary tables individually to the journal.

- Lines 242-243: check grammar.

Thanks to the reviewer for spotting this, the sentence has now been updated to: “Furthermore, genes downregulated in *APOE4* microglia are enriched in the DAM cluster were associated with negative regulation of tumor necrosis (TNF) cytokine production (**Fig. 4b**).”

- Line 261: the evidence that APOE4 microglia acquire a CRM-like state is weak, therefore the statement should be softened.

We thank the reviewer for this suggestion and have made it clearer that this was shown in an independent study using the same xenotransplantation model but with single-cell sequencing: “Mancuso and colleagues (Mancuso et al. 2024) showed that in response to A β , *APOE4* microglia shift to the CRM state rather than HLA.” We have also removed: “, and instead switch to more pathological states such as CRM”.

- Fig. 6D is not referred to in the manuscript.

Thank you to the reviewer for spotting this, we now refer to this figure in the revised manuscript.

- Lines 329-330: check grammar.

Thanks to the reviewer for spotting this, the sentence has now been updated to: “Here, we show that the human *APOE2*, *APOE3*, and *APOE4*, differentially regulate microglia in the context of A β aggregates.”

- Methods: specify what age were the mice when microglia were harvested.

We have now added to the methods section that the mice were 12 months old.

- Discussion: is it accurate to refer to this xenotransplantation model as “late-onset AD” (see DOI: 10.1002/alz.13840), because the pathology is induced via mutations in APP (which cause familial AD)?

We have now removed the term “late-onset” when referring to the AD mouse model from the discussion.

Reviewer #4 (Remarks to the Author):

In this manuscript, Murphy and colleagues use transcriptomics and open chromatin analysis to describe potential differences in gene expression and putative gene regulation between different ApoE isoforms. This work features the use of iPSC cells which have been engrafted into a humanized Alzheimer's disease model; this methodology would allow in vivo functional interrogation of human genetic mutations with enhanced translational applicability outside of traditional mouse models. The authors found that the APOE4 isoform was de-enriched in genes associated with the DAM subset, which has been shown to be protective in mouse AD models. Additionally, APOE4 microglia upregulate pro-inflammatory cytokines, reflecting past mouse studies with humanized APOE4 alleles. However, the authors also found that some selective pro-inflammatory cytokines were upregulated in the APOE2 isoform, which is known to be protective against AD, raising future questions of how the inflammatory status of microglia may have temporal beneficial and antagonistic effects on AD progression. Lastly, the authors also found enrichment for a motif matching the Vitamin D receptor, suggesting the possibility of differential effects of vitamin D signaling dependent on APOE isoforms. This study appears to be an additional follow-up the original humanized mouse engraftment studies completed by Fattorelli et al., Nat. Protocols 2021. The authors made several observations that are in-line with previous publications also investigating APOE isoforms as well as new observations into potential gene regulation mechanisms. Despite using several APOE isoforms, there is a lack of strong inferential conclusions made about the pattern of gene expression or putative mechanisms being changed across the APOE isoforms.

Major Comments:

1. The manuscript would be stronger and more applicable to translational studies if the iPSC microglia were not null in the other allele. These iPSC lines are available in Fattorelli et al. Can the authors explain why only 1 allele was used and the other was null, especially given that APOE4/4 significantly raises risk over APOE4/0.

We chose to use APOE4/0 as the single cell gene expression study by Mancuso and colleagues (2024) showed that there were no transcriptomic differences in this model when using APOE4/4 versus APOE4/0.

2. Are there any inferences that can be made about APOE2 in relation to the previously published single-cell data?

As described in our response to Reviewer 3, genes upregulated in APOE2 microglia (see downregulated in E4vsE2 and upregulated in E2vsE3 in **Fig. 4a**) are enriched within DAM and HLA. This is the opposite to what we see for APOE4 microglia.

3. Are there any differential transcription factor motifs enriched for open chromatin regions that are differential between APOE2 and APOE4 and all of the other comparisons?

We thank the reviewer for bringing this up, we have now included the results for the other APOE comparisons in the supplementary tables and have highlighted an additional interesting enrichment: “In addition to changes to cytokine expression, regions with increased chromatin accessibility in APOE4-expressing microglia were enriched for STAT2, a TF involved in interferon response signalling. Increased interferon signaling as well as upregulation of associated genes has already been shown in mouse microglia expressing *APOE4* (Machlovi et al. 2022).”

4. The manuscript would strongly benefit from reorganization as the premise is a comparison between the isoforms. However, in some sections, some isoforms are mentioned while others are neglected and how the results related to an APOE null, etc. It is highly difficult to synthesize what are the major findings of this manuscript is that is unique from what has already been published.

While the premise was to compare the isoforms, we wanted each section to be based on analysis type. For example, we start by discussing the RNA-seq results. This then ties into looking at which transcriptomic changes are driven by a gain or loss of function using the KO data. As we only had two samples after QC for the APOE-KO in the ATAC-seq data, we did not draw any conclusions here. We systematically undertook analyses for all proposed comparisons (E2vsE3, E4vsE3, E4vsE2) for each analysis type. Not all analyses yielded significant results for all comparisons. For example, in the WGCNA analysis, we performed differential expression analysis between the modules and then characterised these modules. Only the genes in the two modules up in APOE2 were significantly enriched in biological pathways.

Minor Comments:

1. Line 70 – please specify what the “human-specific microglial response” is? The way it is written is ambiguous.

We have rewritten this to say “In addition, shift into what is thought to be a protective and human-specific microglial state was impaired in *APOE4* microglia¹⁶.”

2. Line 80 – typo: Apoe3 is missing

Thanks to the reviewer for spotting this, we have now added APOE3.

3. Figure 1: CD11+, CD4b+ (wrong label)

Thanks to the reviewer for spotting this, we have now fixed it to: CD11+ hCD45+

4. Figure 1: figure legends can use more detail that match the text (e.g. number of samples with what assay)

We have now included the number of samples used (after QC) for each assay: “Experimental design for xenotransplantation of iPSC-derived human microglia into the brains of *App*^{NL-G-F} mice,

followed by ATAC-seq (*APOE2* = 5, *APOE3* = 5, E4 = 4, *APOE-KO* = 2) and RNA-seq (*APOE2* = 5, *APOE4* = 4, *APOE4* = 5, *APOE-KO* = 3).”

5. SFig2 is the same as Fig1b.

Apologies if this wasn't clear but **Fig. 1b** excludes the KO samples that were removed due to having higher than expected APOE expression, **Supplementary Fig. 2** includes all KO samples.

Reviewer #4 (Remarks on code availability):

Code is appropriate; data files are provided and code is appropriately annotated on a figure basis. We would like to thank the reviewer for taking the time and effort to look through the files and code, we know how much work this is! And we are delighted that they have found the repository complete and well annotated.

Responses to the reviewers for NCOMMS-24-44479-T “The APOE isoforms differentially shape the transcriptomic and epigenomic landscapes of human microglia in a xenotransplantation model of Alzheimer’s disease”

We would like to thank the reviewers for offering their time and expertise to review our revised manuscript. Please see our point-by-point response to the reviewers below.

Reviewer #1 (Remarks to the Author):

The authors have addressed all of my original concerns and have significantly improved the manuscript overall. However, the FACS gating strategy that is now included in Suppl. Figure 9 raises new concerns, as it differs from the strategy reported in Mancuso et al., 2024, where CD11b+ cells were gated first and a distinction was then made between mouse CD45+ and human CD45+ cells. Why was a different strategy chosen here? Most concerning, the sorted human CD45+ cells are not clearly positive for CD11b, indicating that the cells analyzed may not (all) be microglia. The authors need to demonstrate that the sorted cells are highly enriched for microglia vs. other immune cell markers to ascertain the identity of the population analyzed.

We apologise to the reviewer for the confusion. The sorting strategy is correct, and we did isolate CD11b+ hCD45+ human microglia for our experiments. The confusion comes from the fact that the system we used (MACS Quant Tyto) in their original version of the software needs the selection of a fluorescent marker (in our case CD11b) that will serve as a threshold (or trigger as Miltenyi calls it) in order to be able to sort. Technically, this comes from the fact that this system is not a classical droplet sorted, and rather captures cells that run through a continuous stream of liquid that is sequentially interrogated by the three lasers in a sequence: violet, blue and red. Given that the distance between each of the interrogation points is physically separated by 50um, the system calculates the speed of each cell (called time of flight by the company) by measuring the time needed for the cell to go from two consecutive detectors and then dividing that by 50um. Once this speed is determined, a gate at the end of the microfluidic chamber will capture the positive event and separate it into a separate sorting chamber. To do this in practice, the researcher needs to determine a channel acting as “trigger” (at time at onset), and a “cell speed” (for time at end). In our case, we defined CD11b as “trigger” and hCD45 as “cell speed”. This effectively means that all the CD11b negative events are thresholded out, and it appears from the plots (as the reviewers well suggested) that the gate included CD11b negative cells. This is the exact same strategy we used for our paper and 2024, with the exception that with this system, one cannot visualize the CD11b negative events once the sorting starts. In the paper by Mancuso et al 2024 we had the chance to transition to a new version of the software that allows us to apply this trigger function based on backscatters (the equivalent to forward scatters in the MACSQuant Tyto system), therefore being able

to display negative events as well. We copy here the link to the Miltenyi website for reference: <https://www.miltenyibiotec.com/DE-en/products/macsguant-tyto.html>

Reviewer #1 (Remarks on code availability):

I have not assessed the code.

Reviewer #2 (Remarks to the Author):

The authors have adequately addressed my concerns. One minor concern remains which relates to Fig. 4. Because of the lower expression level of APOE4 shown in Figure 1b, it is critical to use adequate and consistent statistical methods. Specifically, although the expression levels of chemokines and cytokines in the APOE4 group are generally higher than those in the E2 and E3 groups, the within-group data exhibit high heterogeneity. As such, when comparing the expression levels of specific genes among the APOE2, APOE3, APOE4, and KO groups, as shown in Figures 4c, d, and e, applying FDR correction appears unnecessary. If a one-way ANOVA or the same statistical test as in Figure 1b is applied, are the differences between the groups still statistically significant? As these are related to a major conclusion of this work, it is critical to apply more rigor to these analyses.

We thank the reviewer for their suggestion ensuring our analyses are statistically robust. Using a one-way ANOVA in DESeq2 (likelihood ratio test), the genes in plot 4c,d are statistically significant after multiple testing correction, but *CCL4L2* represented in plot 4e does not reach statistical significance. We chose to use the FDR-corrected values from the DESeq2 analysis as DESeq2 is specifically designed for count data (which follow a negative binomial distribution), accounts for overdispersion of the data, and is better suited to handling between-sample variability.

Reviewer #3 (Remarks to the Author):

We appreciate the authors' response. The authors have gone through all comments and addressed each point.

The two primary pieces of experimental data the authors added were 1) phagocytosis assays by microglia and 2) FACS analysis of cell abundance. Many of the other validation cases were through other's work or referring to past work in Mancuso et al.

Although interesting, the myelin phagocytosis assay is used as a "parallel" to chemotaxis- this seems a bit of a stretch. Phagocytosis and chemotaxis are different things and rely on different pathways. Therefore, using phagocytosis assays to functionally validate chemotaxis does not seem sound.

We thank the reviewer for pointing this out and wanted to clarify that we did not mean for the phagocytosis assay to be used as a parallel for chemotaxis. We hypothesised that phagocytic function may also be enhanced in *APOE2* microglia and indeed found that *APOE2*-expressing microglia genes are overrepresented in gene sets associated with microglia that phagocytose amyloid beta plaques. Following this, we showed enhanced phagocytic activity functionally *in vitro*. We have now updated the order of panels in the figure and the associated section to explain this more clearly:

“Previous studies have linked the *APOE2* isoform to enhanced phagocytic capabilities (Zhao et al. 2009; Castellano et al. 2011; Wang et al. 2022). Using a recently generated scRNA-seq dataset from phagocytic microglia associated with A β plaques⁵², genes upregulated in *APOE2*-expressing microglia when compared to both *APOE3* and *APOE4*, were exclusively overrepresented in a set of genes upregulated in phagocytic microglia (**Fig. 6a**). To functionally validate this finding, we performed phagocytosis assays using iPSC-derived human microglia using pHrodo *E. coli* particles and fluorescent myelin. As hypothesised, *APOE2*-expressing microglia internalised significantly higher amounts of pHrodo *E. coli* compared to all other *APOE* groups (**Fig. 6b-c, Supplementary Fig. 7**). Additionally, a significantly higher proportion of *APOE2*-expressing microglia successfully took up myelin compared to *APOE4*-expressing microglia (**Fig. 6d-e**), as reported previously⁵⁰. We also observed that *APOE*-KO microglia displayed very high amounts of intracellular myelin. This could indicate an impairment to digest the phagocytosed material and is in line with previous studies showing lipid accumulation in ApoE^{-/-} mouse microglia in models of demyelination⁵¹. Overall, the enrichment of proliferation, migration, phagocytosis, and immune responses, suggests enhanced microglial function in *APOE2*.”

Figure 6. iPSC microglia harbouring different *APOE* allelic variants display differences in the uptake of *E. Coli* pHrodo and fluorescent myelin particles. **a** Barplot showing the overrepresentation of genes upregulated in *APOE2* microglia in a set of genes upregulated in phagocytic microglia responsive to A β plaques⁵². The dashed line represents the significance threshold after FDR correction ($p < 0.05$). **b,c** Representative images and quantification of pHrodo *E. Coli* particles (100ug/ml) uptake by *APOE2*, *APOE3*, *APOE4* and *APOE-KO* iPSC-derived microglia. Each data point

represents an independent well. An average of 1236 +/- 87 cells were quantified per well. **d,e** Representative images and quantification of PKH67 labelled myelin (200ug/ml) uptake by APOE2, APOE3, APOE4 and APOE-KO iPSC-derived microglia. Each data point represents an independent well. An average of 211 +/- 9 cells were quantified per well. Statistical analysis was performed using ANOVA, with Bonferroni correction for multiple comparisons; * $p < 0.05$, ** $p < 0.02$ and *** $p < 0.001$.

I realize the difficulty of the xenotransplantation experiments. However, the authors seem to continually refer to Mancuso et al and other pieces of work to support their sequencing datasets. This renders the current paper more of a data resource rather than a study with strong validation and a better fit for another journal.

Reviewer #3 (Remarks on code availability):

Unfortunately, I do not have the expertise to review the code.

Reviewer #4 (Remarks to the Author):

The authors have adequately addressed the comments. The additional phagocytosis data adds to the Vitamin D receptor finding. Clarifications were also made regarding methodology and exclusions.